# Chloroviruses

**DOI:** 10.3390/v12010020

**Published:** 2019-12-23

**Authors:** James L. Van Etten, Irina V. Agarkova, David D. Dunigan

**Affiliations:** Department of Plant Pathology, Nebraska Center for Virology, University of Nebraska-Lincoln, Lincoln, NE 68583-0900, USA; irina@unl.edu (I.V.A.); ddunigan2@unl.edu (D.D.D.)

**Keywords:** algal viruses, *Phycodnaviridae*, giant viruses, NCLDVs, chloroviruses

## Abstract

Chloroviruses are large dsDNA, plaque-forming viruses that infect certain chlorella-like green algae; the algae are normally mutualistic endosymbionts of protists and metazoans and are often referred to as zoochlorellae. The viruses are ubiquitous in inland aqueous environments throughout the world and occasionally single types reach titers of thousands of plaque-forming units per ml of native water. The viruses are icosahedral in shape with a spike structure located at one of the vertices. They contain an internal membrane that is required for infectivity. The viral genomes are 290 to 370 kb in size, which encode up to 16 tRNAs and 330 to ~415 proteins, including many not previously seen in viruses. Examples include genes encoding DNA restriction and modification enzymes, hyaluronan and chitin biosynthetic enzymes, polyamine biosynthetic enzymes, ion channel and transport proteins, and enzymes involved in the glycan synthesis of the virus major capsid glycoproteins. The proteins encoded by many of these viruses are often the smallest or among the smallest proteins of their class. Consequently, some of the viral proteins are the subject of intensive biochemical and structural investigation.

## 1. Algal Viruses

Viruses that infect higher plants are typically small, plus-stranded RNA containing particles that only encode a few genes [1]. Although small viruses have also been discovered that infect eukaryotic algae; e.g., *Marnaviridae* (e.g., Nagasaki [2]; Miranda et al. [3]; Coy et al. [4]), many viruses that infect algae are large dsDNA-containing viruses with genomes ranging from 160 to 450 kb containing up to ~550 predicted protein-encoding genes (CDSs) and many tRNA genes. These large viruses (family *Phycodnaviridae*), which constitute a rapidly growing, genetically and morphologically diverse group of viruses, are found in aqueous environments throughout the world and can play dynamic roles in regulating algal communities such as the termination of massive algal blooms that can be observed from space, commonly referred to as red and brown tides (e.g., Furhman [5]; Suttle [6]; Wommack & Colwell [7]). However, not all large dsDNA viruses that infect algae are actually members of the *Phycodnaviridae* family but are more closely related to the *Mimiviridae* family, and they are in the process of being reclassified by the ICTV [8].

This review focuses on one of the six genera in the *Phycodnaviridae*, genus *Chlorovirus*, that are icosahedral-shaped dsDNA-containing viruses that infect and replicate in certain strains of unicellular, symbiotic, chlorella-like green algae. As noted below, chlorovirus structure, their initial and terminal stages of infection, and some of their genes resemble bacteriophages more than viruses that infect eukaryotes. The lytic chloroviruses can be produced in large quantities and assayed by plaque formation (Figure 1A) using standard bacteriophage techniques [9]. Currently there are over 450 publications on the chloroviruses and their gene products. Prior reviews that focused primarily on the chloroviruses were published in 2012 and 2016 [10,11] and a more personalized description of the discovery of the chloroviruses was published in 2015 [12].

## 2. General Properties of the Chloroviruses

Chloroviruses exist in inland waters throughout the world with titers of a single type occasionally reported as high as thousands of plaque-forming units (PFU) per ml of native water; however, typically chlorovirus titers are in the range of 1 to 100 PFU/mL. Titers fluctuate during the year with the highest titers typically occurring in the spring, followed by another increase in the late fall—at least in a single urban lake in a temperate zone (e.g., Quispe et al. [13])—while in mid-summer there can be few detectable viruses in the water column. Chlorovirus genome sequences have also been detected in metagenomes from marine environments [14] but no chloroviruses have been isolated from these environments. Chlorovirus hosts, which are normally mutualistic endosymbionts and are often referred to as zoochlorellae [15,16], are associated with the ciliate protozoan *Paramecium bursaria* (Figure 1B) [17], the coelenterate *Hydra viridis* [18,19], or the heliozoon *Acanthocystis turfacea* (Figure 1C) [20]. The zoochlorellae live in the ciliate in a special perialgal vacuole that is surrounded by a host-derived membrane, which protects them from attack by host lytic enzymes. In this relationship, the zoochlorellae carry out photosynthesis and release sugars to the paramecia (e.g., Reisser [21]; Shibata et al. [22]), which is a good reason to have zoochlorellae as a partner. In contrast, the reason the zoochlorellae would become a symbiont was unknown. However, the finding that the zoochlorellae are resistant to the chloroviruses in their symbiotic state provides an advantage to the algae.

Fortunately, some zoochlorellae can be grown independently of their partners in the laboratory, permitting plaque assay of the viruses and synchronous infection of their hosts, which allows one to study the virus life cycle in detail. Four such zoochlorellae are *Chlorella variabilis* NC64A and its viruses are called NC64A viruses, *Chlorella variabilis* Syngen 2-3 and its viruses are called Osy viruses [13], *Chlorella heliozoae* SAG 3.83 and its viruses are called SAG viruses [23] and *Micractinium conductrix* Pbi and its viruses are called Pbi viruses [24]. Attempts to culture the zoochlorellae from hydra to serve as a host for the viruses that infect hydra zoochlorellae were unsuccessful [25]. Many “green” organisms are holobionts with zoochlorellae and these represent a large potential for further chlorovirus discovery.

Very little is known about the natural history of the chloroviruses and we suspect that many more chloroviruses and hosts exist in nature (e.g., see Short et al. [26]). Furthermore, chlorovirus DNA sequences are often reported in metagenomic studies from the ocean, which suggests that at least some chloroviruses infect marine algae or some other marine organism(s) (e.g., Flaviani et al. [14]).

Paramecium bursaria chlorella virus-1 (PBCV-1), which infects *C. variabilis* NC64A, is the type member of the genus *Chlorovirus* [27,28]. Typically, species and strains of *Chlorella*, a nonsexual genus, have a simple developmental cycle. (Note, the *C. variabilis* NC64A genome sequence indicates it has many of the genes required for a sexual stage [29].) Vegetative cells increase in size and divide into two, four, eight, or more progeny (called autospores), which immediately develop into vegetative cells. The number of progeny formed from a single vegetative cell depends on the *Chlorella* species and the environmental conditions [30]. *C. variabilis* NC64A cells typically produce four progeny on standard growth medium when grown in constant light at 25 °C. This results in a doubling time of about 24 h for the alga under laboratory conditions.

The taxonomy of the *Chlorella* genus has an interesting history in that originally many small spherical or ellipsoidal, unicellular, nonmotile, asexually reproducing green algae were assigned to the genus. Most *Chlorella* species are free-living and only a few of them live as hereditary endosymbionts. *Chlorella* species have a rigid cell wall and typically have a single chloroplast, often containing a pyrenoid body. To illustrate the situation in chlorella taxonomy prior to molecular taxonomy, the nuclear DNAs of algae originally assigned to the *Chlorella* genus had G+C values ranging from 43% to 79% [31]. Huss et al. (see [32] and references cited therein) also examined the relationships among algae originally classified as *Chlorella* species by using DNA base ratios, DNA hybridization, and DNA re-association kinetics to classify the algae. Currently many of the algae originally assigned to the *Chlorella* genus are being reclassified by sequencing their 18 S ribosomal DNAs (e.g., Darienko et al. [33]).

The dogma is that the four groups of chloroviruses each infect only one host. However, there is one report that NC64A viruses can also infect and replicate in another *Chlorella* species named *C. prototechoides* 211-6 [34]. This alga was reported to have been isolated as a zoochlorellae from *Paramecia bursaria* in the United States in 1936. Subsequent physiological properties and pulsed-field gel electrophoresis experiments established that *C. variabilis* NC64A and the *C. prototechoides* 211-6 isolate differ. To our knowledge the *C. prototechoides* 211-6 host for the NC64A viruses has not been studied further.

Early studies reported that PBCV-1 could replicate normally in cells that had either been exposed to the photosynthetic inhibitor 3-(3,4-dichlorophenyl)-1,1-dimethylurea or in cells grown in the dark [35]. However, PBCV-1 infection rapidly reduced photosynthetic activity by the host [36]. Like many viruses, PBCV-1 replicates better in actively growing cells than in cells that have reached stationary growth [35]. This suggests that there is a source of metabolic energy other than from photosynthesis.

Surprisingly, PBCV-1 replicates slowly and with a smaller burst size in UV-irradiated cells that can no longer form colonies, which would suggest that the virus does not require labile host factors for replication [37]. These experiments were conducted with cells maintained in the dark after exposure to light to avoid photo-induced DNA repair and were conducted prior to the discovery that the chloroviruses encode a DNA repair enzyme that is expressed shortly after infection [38]. The role that this virus-encoded DNA repair enzyme plays, if any, in the UV experiments described above is unknown.

PBCV-1 infectivity is rapidly destroyed by exposure to chloroform and more slowly with exposure to ethyl ether or toluene. In contrast, detergents such as 2% Triton X100, 2% Nonidet P-40, and 2% sodium deoxycholate have no effect on viral infectivity. However, PBCV-1 slowly loses infectivity in 2% sodium N-lauryl sarcosine and rapidly in 2% sodium dodecyl sulfate at room temperature [39].

Given the coding capacity of the chloroviruses, it is not surprising that they encode many unexpected putative proteins. However, with the exception of homologs solely in other chlorovirus members, about 50% of their CDSs do not match anything in the databases. Some chloroviruses also have as many as three types of introns and at least one NC64A virus, NY-2A, has two inteins, one in the large subunit of ribonucleotide reductase and one in a superfamily II helicase [40].

The *C. variabilis* NC64A 46.2 Mb genome has been sequenced and is predicted to code for ~9800 proteins [29]. The availability of both host and virus sequences makes chloroviruses an attractive model system to study. Recently, the *M. conductrix* genome, the host for the Pbi viruses, was sequenced and it is predicted to code for ~9400 proteins [41].

## 3. PBCV-1 Structure

The Mr of PBCV-1 is ~1 × 10^9^ Da and the S_20,w_ value is ~2300 S. The extinction coefficient for PBCV-1 is A^260^_0.1%_ (uncorrected for light scatter) 10.7 ± 1 [35]. Some chlorovirus particles are disrupted in CsCl and also by freeze-thawing. Chloroviruses are generally stable in sucrose but if not, they are stable in iodixanol. The virion consists of ~65% protein, 5%–10% lipid, and ~25% DNA on a weight basis [39]. The predominant lipids are phosphatidyl choline, phosphatidyl ethanolamine, and an unidentified component [39]. Low concentrations of EDTA and EGTA have no effect on PBCV-1 infection at room temperature. Treatment of PBCV-1 with reducing agents produce interesting results. Exposure to either 5 mM dithiothreitol or 5 mM dithioerythritol results in the rapid loss of PBCV-1 infectivity. In contrast, 5 mM mercaptoethanol has no effect on PBCV-1 infectivity [39]. In the original purifications of PBCV-1 we noted that filtration of the virus through 0.4 µm Millipore filters led to disruption of the virus and loss of infectivity. In contrast, filtration through 0.4 µm Nucleopore filters had no effect on virus infectivity [42].

Proteomic analysis of highly purified PBCV-1 virions indicate that the virus contains at least 149 different proteins of which 148 are virus-encoded and range in size from <10 to >200 kDa [43]. The 148 virus-encoded proteins represent ~35% of the coding capacity of the virus. Some viral proteins appear to be structural, whereas others have enzymatic, chromatin modification, and signal transduction functions. Most (106) of the viral-associated proteins have no known function or homologs in the existing gene databases except as orthologs with proteins of other chloroviruses, phycodnaviruses, and nuclear-cytoplasmic large DNA viruses (NCLDV). The PBCV-1 genes encoding these proteins are located throughout the virus genome and most are transcribed either early/late or late in the infectious cycle, which is consistent with virion morphogenesis. At least one virion-associated protein is host-encoded (a predicted DNA-binding protein). PBCV-1 virions have many predicted transmembrane proteins and several major capsid-like proteins. In addition, the PBCV-1 virion has three glycoproteins, three myristylated proteins and several phosphoproteins [44,45]. Some of the virus-encoded proteins that have been identified in the PBCV-1 particle include homospermidine synthase, a histone 3–Lys^27^ methylase, a prolyl-4-hydroxylase, a Cu/Zn superoxidase dismutase, two DNA restriction endonucleases, and several protein kinases.

The PBCV-1 major capsid protein (MCP) is a glycoprotein with a molecular weight of ~54 kDa; the peptide portion of the protein is 48.2 kDa [49] with sugars contributing the remaining molecular weight. The MCP consists of two eight-stranded, antiparallel β-barrel jelly-roll domains (Figure 2A). The monomeric MCP forms a trimeric capsomer (Figure 2B), which has pseudo-sixfold symmetry (Figure 2C) [50]. It is interesting that the MCP migrates on SDS-PAGE at ~54 kD after heating to 100 °C; however, after heating to 60 °C, the MCP migrates at about 110 kD. Therefore, it was originally thought the MCP existed as a dimer. However, it is now known that it exists as a trimer in the viral capsomers with a size of ~162 kD.

The structures of the four Asn-linked glycans attached to the PBCV-1 MCP have been solved; they consist of 8–10 neutral monosaccharide residues, producing a total of four glycoforms (Figure 3) [53]. These glycan structures are unique and do not resemble any structure previously reported in the three Domains of Life. Some of their most distinctive features include: (i) the four glycoforms share a common core structure; (ii) The glycoforms are related to the non-stochiometric presence of two monosaccharides, l-arabinose and d-mannose; (iii) The most abundant glycoform consists of nine neutral monosaccharide residues organized in a highly-branched fashion (Figure 3); (iv) None of the N-linked glycans are attached to a typical Asn-X-(Thr/Ser) consensus site in the protein; (v) The glycans are attached to the protein by a β-glucose linkage, which is rare in nature and has only been reported in glycoproteins from a few organisms [54,55,56,57]; (vi) The glycoform contains a dimethylated rhamnose as the capping residue of the main chain, a hyper-branched fucose residue and two rhamnose residues with opposite configurations.

The structure of these n-glycans consists of two regions: the core region, consisting of four sugars, is located near the protein backbone and is conserved among the chloroviruses [58]. The core consists of an Asn-linked glucose, which is linked to a terminal xylose unit and a hyperbranched fucose, which is in turn substituted with a terminal galactose. The third position of the fucose unit is always linked to a rhamnose, which is a semi-conserved element because its absolute configuration is virus-dependent. The second region extends the conserved core with other monosaccharides, which are specific for each chlorovirus [58]. A detailed description of the complete structure of the PBCV-1 MCP was recently described by combining crystallographic data with molecular modeling (Figure 2A) [51]. This is one of the first glycoproteins, if not the first, to have its complete structure resolved. As elaborated on below, the chloroviruses are unusual in that they encode most if not all of the machinery to glycosylate their MCPs.

We originally reported that the PBCV-1 MCP had a myristic acid attached to an internal portion of the protein [44]. However, we have been unable to confirm the presence of myristic acid in subsequent structural studies on the PBCV-1 MCP, even though the original experimental results were very convincing. The reason for this discrepancy is unknown. We also originally reported that the MCP had two O-linked glycans [50] but subsequent studies indicate that that conclusion was incorrect [51].

The two eight-stranded, antiparallel β-barrel jelly-roll domains in the PBCV-1 MCP occur in the MCPs of viruses that infect organisms in all three Domains of Life [59]. This finding led to the suggestion that viruses from each of these domains might be evolutionarily related even though the amino acid sequences of their MCPs differ substantially [59,60].

Cryo-electron microscopy (cryoEM) and six-fold symmetry averaging three-dimensional reconstruction of PBCV-1 virions at ~26 Å resolution indicated that the capsid was an icosahedron that is 1900 Å in diameter from point-to-point with a triangulation number of 169d [52]. The capsid covers a single lipid bi-layered envelope, which is required for infection [52,61,62]. The trimeric capsomers are arranged into 20 triangular facets (trisymmetrons, each containing 66 capsomers) and 12 pentagonal facets (pentasymmetrons, each containing 30 trimers and one pentamer at each of the icosahedral vertices) (Figure 4A). However, as cryoEM procedures have improved, more details have emerged about the structure of the PBCV-1 virion; the virus is much more complex than the original icosahedral structure suggested.

Improvement to 8.5 Å resolution and five-fold symmetry 3D reconstruction of cryoEM images revealed that one PBCV-1 vertex has a 560 Å long spike-structure (Figure 4A); 340 Å extends from the surface of the virus [61,62]. The part of the spike structure outside the capsid has an external diameter of 35 Å at the tip, expanding to 70 Å at the base. The spike structure widens to 160 Å inside the capsid and forms a closed cavity inside a large pocket between the capsid and the membrane enclosing the virus DNA (Figure 4B). Therefore, the internal virus membrane departs from icosahedral symmetry adjacent to the unique vertex (Figure 4B). Consequently, the virus DNA is packaged non-uniformly in the particle. (This asymmetric packaging of the genome was reported previously in an ultra structural study of the Pbi virus CVG-1 [63] and is apparent in some other earlier electron microscopy studies). External “fibers” extend from some of the trisymmetron capsomers (one per trisymmetron) and are rigid enough to be resolved with cryo-EM methods; the fibers presumably aid in virus attachment to the host (Figure 4A,C). The fiber-containing capsomer is always located in the middle of the second row of capsomers in the trisymmetron.

Recently, a 3.5 Å near-atomic resolution structure of PBCV-1 by cryoEM was reported [64]. This resolution was accomplished by correcting for the Ewald sphere effect in single cryoEM reconstructions [65]. Otherwise, the technology was stalled at ~4.5 Å resolution. The 3.5 Å resolution led to the identification of 14 virus-encoded, minor capsid proteins that form a hexagonal network below the outer capsid shell, stabilizing the capsid by binding neighboring capsomers together (Figure 4D). The protein located at 11 of the 12 virus vertices was identified as A310L (labeled P1 in Figure 4D). The penton protein folds into a canonical jelly-roll structure [66] composed of eight β-strands arranged into the standard two four-stranded β-sheets that are stabilized primarily by their hydrophobic interfaces. A310L is a glycoprotein containing 4 N-linked glycans that, like the MCP, are not located at the canonical Asn-X-(Thr/Ser) sequon recognized by endoplasmic reticulum (ER) located glycosyltransferases. We predict that the glycan structures attached to A310L will resemble the glycan structures associated with the virus MCP.

The size of the viral capsid is predicted to be determined by a “tape measure” minor capsid protein (protein P2 (A342L) in Figure 4D) [64]. Currently, research efforts are focusing on identifying the proteins associated with the spike structure as well as the proteins involved in forming the fibers.

One long-standing question about the chloroviruses and other large icosahedral viruses is: how are the viruses assembled? Initially one assumed that the capsomers were first assembled into trisymmetrons and pentasymmetrons and then joined together to form the capsid. Consistent with this concept is that within individual trisymmetrons the capsomers are packed in the same orientation, but between neighboring trisymmetrons the orientation of the capsomers differs by 60 degrees. However, one problem with this concept is that scientists who have studied electron micrographs of thin sections of cells infected with large icosahedral viruses, including PBCV-1, have never seen patches of capsomers resembling independent trisymmentrons or pentasymmetrons inside host cells. This makes one suspect that they are not synthesized independently and then joined together. Furthermore, recent electron tomography studies of thin sectioned cells infected by either vaccinia virus or minivirus suggest that the assembly starts from one 5-fold vertex and seems to be continuous [67]. These observations have led to the hypothesis that electrostatic interactions in the capsids are responsible for assembly of PBCV-1 and other large icosahedral viruses [68]. This concept is consistent with the observation that the capsid structures assemble around the internal membrane, i.e., the membrane appears to serve as the template for forming the virus particles (Figure 1H), [47,48].

## 4. Chlorovirus Genomes

The chloroviruses have linear genomes that are 290 to 370 kb in length, which code for ~330 to ~415 CDSs. The termini of the linear 331 kb, non-permuted chlorovirus PBCV-1 genome consist of 35 nucleotide-long, covalently closed hairpin loops that exist in one of two forms. The two forms are complementary when the 35-nucleotide sequences are inverted [69]. Each hairpin loop is followed by an identical 2.2 kb inverted repeat sequence [70]; the remainder of the genome consists primarily of single-copy DNA. The PBCV-1 genome was re-sequenced in 2012 to correct mistakes in the original sequence that was reported in the mid-1990s ([71] and references listed therein). Using 40 codons as the minimum CDS size and avoiding large overlaps, PBCV-1 contains 416 CDSs (three outermost rings in Figure 5) [43], (GenBank JF411744). Of the predicted CDSs, ~50% resemble proteins of known function, including many that are novel for a virus. The CDSs are distributed nearly evenly on both DNA strands with minimal intergenic spaces. The one exception is a 1788-nucleotide sequence near the middle of the PBCV-1 genome that encodes 11 tRNAs. A more comprehensive study of chlorovirus CVK2-encoded tRNAs revealed that CVK2 encoded 14 clustered tRNAs with intergenic spacers of 1–30 bp, which were co-transcribed as a large precursor and then processed to mature tRNAs [72]. However, none of the tRNAs have a CCA sequence at the 3′ end of the acceptor stem of the tRNAs. Like other cellular tRNAs, these 3 nucleotides are added separately to the tRNAs by a host enzyme; the tRNAs were also shown to be aminoacylated in vivo [72]. Typically, the chlorovirus-encoded tRNAs support the codon usage necessary for a shift from the 67% G+C genome of the host cells to the 40% G+C content of the viral genomes for the NC64A viruses (e.g., Lee et al. [73]). However, individual chloroviruses encode different tRNAs. Many of the core protein genes are in conserved clusters and are referred to as “Gene Gangs” [74]. Unexpected patterns in the genomic structure were observed using a set of 41 *Chlorovirus* genomes, which come from three phylogenetic clades where the members of each clade share the same host algal strain (Pbi, SAG, and NC64A viruses). The identification of conserved gene structure is important for predicting if these features are under selective pressure, and therefore likely to be biologically relevant, as well as advancing the study of genome organization and evolution. Applying this analysis to the chloroviruses was an avenue to gain insight into mechanisms of genome evolution in this virus group, and possibly in large dsDNA viruses more generally. Qualitative assessment of pairwise genome alignments suggested homologous genes tend to be localized together in chlorovirus genomes and more so within viral clades. This co-localization does not involve strict conservation of gene order or strandedness. Further quantitative testing of the constraint in the distances between homologs present in all genomes indeed indicates a small fraction of homologous genes are more likely to be found within <5 kb of each other than expected by chance. Conservation in homolog pair distances is larger (~15 kb) in genomes of the same viral type. By extending the search for gene clusters outside of homologs identified as “Gene Gangs”; i.e., groups of three or more genes that are co-localized on the genome in the majority of chlorovirus genomes, the majority of genes in gangs are homologs. The presence of Gene Gangs indicates there is selection for functionally related genes to remain in proximity, helping to attribute functional annotations to otherwise uncharacterized genes, and implies the existence of novel regulatory mechanisms that coordinate their expression.

The coding regions of some of the genes slightly overlap and early and late genes are dispersed throughout the PBCV-1 genome, although there is some clustering of the early virus genes [75]. Not all PBCV-1 genes are required for virus replication in the laboratory. Four spontaneously derived PBCV-1 mutants were isolated that contain 27- to 37-kb deletions at the “left” end of the genome [76]. Collectively, ~40 kb of single-copy DNA encoding 31 CDSs, or ~12% of the PBCV-1 genome, is not required for PBCV-1 replication in the laboratory, although its replication is attenuated. The isolation of large deletions in another chlorovirus, CVK-1, was reported at the same time as the PBCV-1 results [77]. Deleted CDSs in PBCV-1 included two putative capsid-like proteins, a putative d-lactate dehydrogenase, a glycerophosphoryldiesterase, two glycosyltransferases (see below), and a pyrimidine dimer-specific glycosylase. Interestingly, the deletion mutants that were missing a significant portion of the “left” end of their genomes, including the terminus, moved a multi-kb duplicate copy of the “right” end of the genome to the “left” end of the genome, thus maintaining the hairpin end and inverted repeat structure [76]. Therefore, the chlorovirus genome structures are somewhat fluid. Interestingly, poxviruses and African Swine Fever virus do something similar in regards to gene deletions at one termini and replacement with a terminal sequence from the other end of the genome (e.g., Turner and Moyer [78]; Vydelingum et al. [79]). Since both groups of viruses have genomes with hairpin ends, we postulate that the hairpin ends are required for the replication of the viral genomes.

The chlorovirus genomes contain methylated bases [80]. Genomes from a selected set of 37 NC64A chloroviruses have 5-methylcytosine (5 mC) in amounts ranging from 0.1% to 48% of the total cytosines. In addition, 24 of the 37 viral DNAs contain N^6^-methyladenine (6mA) in amounts ranging from 1.5% to 37% of the total adenines [27]. The chlorovirus genome with the most methylation is NC64A virus NY-2A, which has 45% of its cytosines and 37% of its adenines methylated. Not surprisingly, its DNA is resistant to cleavage by most known restriction endonucleases. The methylated bases in the chloroviruses occur in specific DNA sequences, which led to the discovery that chloroviruses encode many 5mC and 6mA DNA methyltransferases. For example, the aforementioned NY-2A virus encodes 18 predicted DNA methyltransferases [81,82]. Some of the chlorovirus encoded DNA methyltransferases recognize very short sequences; e.g., one enzyme methylates cytosine in GpC sequences and it is useful for chromatin studies in vivo [83]. Nothing is known about methylation in the SAG virus genomes.

About 25% of the virus-encoded DNA methyltransferases have companion DNA site-specific (restriction) endonucleases, which typically recognize short (2 to 4 bp) sequences. Some of the enzymes with unique cleavage specificities are sold commercially (e.g., Van Etten et al. [27]; Nelson et al. [84,85]; Chan et al. [82]). One of the restriction endonucleases, CviJI, is interesting because under certain conditions (referred to as CviJI*) it cleaves PuGCPy, PyGCPy, and PuGCPu sequences between the G and C, thus producing blunt ended DNA fragments. Consequently, CviJI* has been used for shotgun cloning of DNA because it requires less DNA and avoids DNA end repair processes [86,87,88]). Two of the virus-encoded DNA endonucleases are unusual because they cleave one strand of the duplex DNA in a site-specific manner, i.e., they are nicking enzymes, Nt.CviPII (/CCD) [89] and Nt.CviQXI (R/AG)[81]. Interestingly, Nt.CviPII does not cleave DNA into fragments smaller than about 25 nucleotides, even though internal cleavage sites exist. Thus Nt.CviPI can create primers from an autonomous spot of DNA that can then be amplified by PCR [90].

The purpose of the virus-encoded DNA methyltransferases, without companion restriction endonucleases [referred to as orphan DNA methyltransferases], is unknown. One possibility is that they were originally associated with companion DNA restriction endonucleases that have subsequently been lost in evolution. Support for this possibility is provided by the observation that chlorovirus PBCV-1 has probably lost a restriction endonuclease encoding gene but it still has its companion DNA methyltransferase; however, the DNA methyltransferase is no longer functional, i.e., it is now a pseudogene [91]. Of course it is also possible that the orphan DNA methyltransferases may serve some other unknown physiological function.

Prior to the discovery that the chloroviruses encoded enzymes with DNA restriction endonuclease activity, all known DNA restriction endonucleases were from bacteria. The prevailing dogma is that these enzymes help prevent phage infection and foreign DNA from entering the cell (e.g., Wilson [92]). However, the chlorovirus situation is unusual because the viruses encode the enzymes and the process occurs in a eukaryotic cell. However, like bacteria, the genes encoding the chlorovirus DNA methyltransferases and DNA restriction endonucleases are always located adjacent to one another, a type of two-component Gene Gang.

About 45 additional chlorovirus genomes have been sequenced and annotated, including members infecting all four host algae [93,94]. About 150 of the genes are common to all of the viruses, suggesting that they are important for virus replication. Eleven, nine, and six genes are only present in the NC64A, SAG, and Pbi viruses, respectively.

The sum total of chlorovirus-encoded CDSs from 45 analyzed viruses (NC64A, Pbi, SAG viruses) include ~650 protein families. Any one virus encodes no more than ~400 CDSs, thus the chloroviruses have a lot of genetic diversity. Furthermore, some chlorovirus-encoded CDSs have as many as three distinct functional domains, and each domain encodes an independent enzyme activity. Therefore, the genetic diversity of the chloroviruses is much larger than the total number of genes. Why chloroviruses need this large amount of gene diversity and the origins of many of these genes is unknown.

Not surprisingly, homologs from viruses infecting the same host are the most similar; e.g., the average amino acid identity between homologs from PBCV-1 and the related NC64A viruses NY-2A and AR158 is ~73%. However, PBCV-1 and Pbi viruses, e.g., MT325 and FR483, orthologs have ~50% amino acid identity, and PBCV-1 and SAG virus ATCV-1 orthologs have ~49% amino acid identity. Using PBCV-1 as a model, there is good synteny among NC64A viruses with only a few readily identifiable localized rearrangements, including inversions and indels. Likewise, the Pbi and SAG viruses exhibit good gene synteny among themselves. On the other hand, NC64A viruses have only slight synteny with the Pbi and SAG viruses (Figure 6) [74,93,95]. The one exception to these statements is the Pbi virus NE-JV-1 that has almost no gene co-linearity with the other Pbi viruses (or other virus types) and is clearly a phylogenetic outlier.

It is interesting that gene order in the poxviruses and mimiviruses is conserved toward the center of their genomes [96,97], whereas significant disruptions of gene colinearity occurs at the genome extremities. In contrast, no obvious differences were observed in the levels of conservation between the center and extremities of the chlorovirus genomes, which suggests a possible different mechanism of genome evolution in the chlorovirus clade [93].

As described above, the analyses for the discovery of Gene Gangs were stimulated by pair-wise comparisons of genomes of the same type (Figure 6A) versus comparing genomes between types of virus (Figure 6B). When tracing all homologs between the three types of viruses, it became apparent that most of the homologs do not follow a pattern (Figure 6C, gray lines). However, a smaller set of homologs can be seen clustered, regardless of the virus type (Figure 6C, color lines). Large scale analyses confirmed that the clustering is conserved. Given the evolutionary “noise” observed with the bulk of the viral genes due to genome plasticity, Gene Gangs may represent multi-gene functionalities that are selected in chloroviruses, for example the five-member Gene Gang 12 appears to be involved in DNA replication, where the gang is composed of two DNA polymerase type-B delta proteins, a PCNA protein, a sine oculis-binding protein, and a hypothetical protein. Some of the Gene Gangs are this transparent, others are not as obvious [74].

The G+C content of the PBCV-1 genome, and the genomes of all the NC64A viruses, is ~40; by contrast, its host *C. variabilis* NC64A nuclear genome is ~67% G+C [80]. The genomes of the Pbi and SAG viruses are ~45% and 50% G+C, respectively.

## 5. PBCV-1 Life Cycle

### 5.1. Virus Entry

Three-dimensional reconstruction of PBCV-1 in the presence of *C. variabilis* NC64A cell walls supports the hypothesis that the virus spike structure first contacts the cell wall (Figure 1D) [62] and that the fibers appear to aid in holding the virus to the wall (Figure 4C) [27]. The PBCV-1 protein A140/145R was reported to be involved in recognizing the host receptor [98] and to be located at a unique vertex [99]. These experiments were conducted prior to the finding that the virus has a spike structure at one vertex. The virus spike structure is too narrow to deliver DNA into the host and likely serves to puncture the wall using a virus-associated enzyme(s) (Figure 1E) before it is pushed aside. PBCV-1 attachment to its host receptor is rapid and specific with an adsorption rate of 5 × 10^−9^/mL/min [46]. The inability to attach to non-host cells is a major factor in limiting chloroviruses’ host range. We were surprised to find that early stages of PBCV-1 attachment are reversible [100]. The identity of the host receptor, which is uniformly present over the entire surface of the alga, is unknown but circumstantial evidence suggests that it is probably a carbohydrate [101]. As an aside we once asked if the zoochlorellae chlorovirus receptor was the same as the zoochlorellae receptor to form a symbiotic relationship with the paramecium by conducting the following experiment. The zoochlorellae host for the Pbi viruses originally came from a paramecium isolated in Europe and the zoochlorellae host for the NC64A viruses came from a paramecium isolated in the United States. Whereas the viruses could distinguish the two zoo chlorellae, the paramecia could not [102]. Therefore, the viruses and the paramecia are recognizing different receptors.

PBCV-1 attaches equally well to isolated walls even if they have been exposed to 100 °C in 2% SDS, extracted with methanol or 2M LiCl or exposed to several proteases. Not surprisingly, PBCV-1 does not attach to *C. variabilis* NC64A protoplasts.

PBCV-1 encodes five proteins that degrade potential cell wall polysaccharides including two chitinases [103], a chitosanase [103], a β-1,3 glucanase [104], and an alginate lyase-like enzyme [105,106,107]. Recombinant proteins have been produced from all of these genes and shown to have the predicted enzymatic activities. It is interesting that a homolog of one of the two PBCV-1 chitinases has been described in chlorovirus CVK2 that has two functional domains. One domain generated chitobiose from chitin and the second domain generated N-acetylglucosamine from chitobiose [108]. The discovery that PBCV-1 encoded three functional enzymes involved in chitin degradation was unexpected because chitin was not known to be present in cells walls of green algae. However, chitin was reported to be present in the cell walls of the host for the Pbi viruses, which at the time was considered to be a *Chlorella* species [109,110]. Molecular taxonomy has now placed the host for the Pbi viruses in the genus *Macractinium* [111]. We predict that chitin also exists in cell walls of authentic *Chlorella* species. In fact, one hypothesis is that components of chlorella chitin metabolism were acquired from the chloroviruses [29].

Proteomics analysis of the PBCV-1 virion did not detect any of these five polysaccharide-degrading enzymes in the virion, as was hypothesized [43]. Therefore, the 148 CDSs coded by the virus that comprise the virus particle were re-examined and this led to the discovery that one of them, A561L, has a domain that resembles an alginate lyase. A recombinant protein was made containing this domain (named A561L^T^, T for truncated) and it has *C. variabilis* NC64A wall degrading activity [112]. Antibody prepared against A561L^T^ reacts with the protein from PBCV-1 and the antibody inhibits the cell wall degrading activity isolated from PBCV-1 virions. Therefore, A561L^T^ is believed to be solely responsible for degrading the cell wall during the entry phase of PBCV-1 infection.

The PBCV-1 encoded β-1,3 glucanase enzyme is interesting because the protein is expressed very early in infection and disappears from the cell by 60 to 90 min p.i. [104]. One possibility is that the enzyme might be used to degrade putative storage β-1,3 glucans in the cell early during infection. If indeed β-1,3 glucans serve as a storage polysaccharide for *C. variabilis* NC64A, this would be a rapid way for the cell to generate glucose for energy during virus infection, which coincides with the suppression of host protein synthesis.

Following host cell wall degradation by the virus-associated A561L^T^ (Figure 1F), the virus internal membrane fuses with the host membrane, forming a membrane-lined tunnel between the virus and its host (Figure 7A,B) [113], leaving an empty capsid attached to the surface [46]. The membrane-lined tunnel between the virus and host is so narrow that the compacted DNA in the virion must pass into the host in a linear manner and then once inside the host it appears to condense again. This DNA condensation may aid movement of the genome to the host nucleus. (Note: unlike many bacteriophage that release their DNA when they come in contact with the host receptor, PBCV-1 does not release its genome when it contacts isolated host cell walls and digests the cell wall.)

The virus-host membrane fusion process triggers rapid depolarization of the host plasma membrane, probably initiated by a virus encoded K^+^ channel (named Kcv for K^+^ channel from chlorovirus) that is located in the virus internal membrane [114]. However, clearly other channels located in the host membrane need to be activated to produce the depolarization [115]. This hypothesis is supported by the fact that infection by all chloroviruses is inhibited by the K^+^ channel blockers barium, and also cesium for some viruses [116]. The depolarization results in the release of K^+^ from the cell [117]. [Note, it is important to mention that two of the more than 60 chloroviruses that have been examined lack an intact Kcv gene. Events associated with infection by these viruses need to be determined.]

The rapid loss of K^+^ from the host and associated water fluxes significantly reduce the host turgor pressure, which aids ejection of viral DNA and virion-associated proteins into the host [115]. Host membrane depolarization also inhibits many host secondary transporters [118] and prevents infection by a second virus [119]. Ejection of PBCV-1 DNA into the host is also aided by the pressure of the condensed virus DNA (ca. 0.2 bp nm^−3^ [120]). This pressure can occasionally be observed when infection of an algal cell by an individual virus particle (virus DNA labeled with a fluorescent stain) fails and the viral DNA is dynamically ejected from the capsid. Assuming that all of the chloroviruses have the same size capsid, the DNA packaging densities of the chloroviruses must differ because the genome sizes can vary from 280 kb to 370 kb.

### 5.2. Virus Replication—Early Phase

None of the chloroviruses encode a recognizable DNA-dependent RNA polymerase (DdRp) gene, and circumstantial evidence plus microscopic images (Figure 7C,D) [113] indicate that PBCV-1 DNA and viral-associated proteins quickly move to the nucleus and commandeer the host transcription machinery. The rapid initiation of virus transcription implies that some component(s) must facilitate active transport of the virus genome to the nucleus. In this immediate-early phase of infection, host transcription rates decrease and the host transcription machinery is reprogrammed to transcribe viral DNA. Some early viral transcripts are synthesized in the absence of de novo protein synthesis [121]. Details on the reprogramming are unknown. However, host chromatin remodeling is probably involved because PBCV-1 encodes and packages an enzyme (referred to as vSET) that tri-methylates Lys-27 in histone 3 [122]. Circumstantial evidence indicates that this methylation inhibits host transcription following PBCV-1 infection [123]. In addition, host chromosomal DNA degradation begins within 5 min p.i., presumably by the PBCV-1 encoded and packaged DNA restriction endonucleases [124]. This degradation of host DNA also aids inhibition of host transcription and release of the host transcription machinery for the benefit of the virus. Chloroplast DNA is also degraded beginning about one h p.i. [125].

PBCV-1 dominates the infected cells in the early phase prior to initiation of viral DNA synthesis, which begins at 60 to 90 min p.i. Microarray experiments [126] indicated that: (i) 98% of the PBCV-1 protein-encoding genes are expressed in laboratory conditions; (ii) 63% of the genes are expressed before 60 min p.i. (early genes); (iii) 37% of the genes are expressed after 60 min p.i. (late genes); and (iv) 43% of the early gene transcripts are also detected at late times following infection (early/late genes). Many of the late and early/late genes encode virion-associated proteins, consistent with particle assembly and maturation.

Subsequently, PBCV-1 transcription was monitored by RNAseq [75], which is a more sensitive procedure than microarrays. RNAseq experiments revealed that ~50 PBCV-1 genes are expressed within the first 7 min p.i. (Figure 5). Infection of *C. variabilis* NC64A by chlorovirus CVK2 also resulted in expression of 23 genes by 5 to 10 min p.i. [127]. By 60 min p.i., essentially all of the PBCV-1 genes are expressed at some level and ~40% of the poly (A+) containing RNAs in the infected cell are PBCV-1 transcripts. This rapid increase in viral mRNAs probably involves the selective degradation of host mRNAs, by an unknown mechanism, together with increased viral transcription. Not surprisingly transcription of host genes is altered within the first 7 min after PBCV-1 infection [128]. As expected, the synthesis of late viral transcripts requires translation of early virus genes.

### 5.3. Transcriptional Control

Consensus promoter regions for early and late PBCV-1 genes have not been identified definitively; however, the sequence AATGACA is common in the 100 nucleotides preceding the ATG start codon of most early PBCV-1 genes [95]. Furthermore, the 50 nucleotides preceding the ATG start codons are usually >70% A+T [129]. Transcription of some PBCV-1 genes appears to be complex. For example, some gene transcripts exist as multiple bands and these patterns can change between early and late times in the virus life cycle.

The promoter region of some of the chlorovirus genes has been tested for promoter activity in both eukaryotic and bacterial cells. For example, a promoter region from an adenine DNA methytransferase gene works as well or better than the cauliflower mosaic virus 35S promoter in transient transformed tobacco, wheat, rice, maize, sorgum, and *Arabidopsis* cells [130,131]. Furthermore, the same promoter worked well in several *E. coli* strains and in some other bacteria [131]. Interestingly, this same promoter and the promoter regions from some of the other chlorovirus genes did not work in the green alga *Chlamydomonas* [132]. In fact, the promoters seemed to inhibit transcription in *Chlamydomonas*. In some ways, it was like the alga knew that the promoter came from an algal infecting virus. However, another chlorovirus promoter has been reported to successfully transform some *Chlorella* species [133,134].

The TTTTTNT transcriptional termination motif present in early vaccinia virus genes [135] frequently occurs 250 to 450 nucleotides downstream of both early and late PBCV-1 ORFs [129] and may play a role in transcriptional termination in the chloroviruses.

PBCV-1 encodes several proteins that are predicted to be involved in virus transcription at some point in the viral life cycle. This includes the following genes that are expressed early: putative transcription factors TFIIB, TFIID, TFIIS, VLTF2, and VLTF3, two helicases (a SWI/SNF helicase and a superfamily II helicase), a RNase III [136] and two proteins involved in capping mRNAs, the smallest known metal-dependent RNA triphosphatase [137] and among the smallest known guanylyltransferases [138,139]. Typically, three enzymatic functions are required to cap the 5′ end of mRNAs that include the two chlorovirus encoded enzyme activities mentioned above plus a methylating enzyme referred to as S-adenosylmethionine: RNA (guanine-N7) methyltransferase. PBCV-1 differs from the well-studied poxviruses, as well as mimivirus, which encode a single protein that carries out all three activities [140,141], but PBCV-1 more closely resembles the process in yeast.

### 5.4. Virus Replication—Late Phase

Host DNA synthesis is inhibited immediately after PBCV-1 infection and viral DNA synthesis begins at ~60 min p.i. [125]; presumably virus DNA synthesis begins in the nucleus before moving to the cytoplasm. By 4 h p.i., the amount of virus DNA in the cell is at least four times higher than the DNA in the cells at the time of virus infection. Therefore, the infected cell must be synthesizing lots of nucleotide precursors, although some of these intermediates presumably come from a salvage pathway of degraded host RNA and DNA (see below). PBCV-1 encodes several enzymes that are involved in viral DNA replication including an α-like (Family B) DNA polymerase [142], a DNA primase, an ATP dependent DNA ligase [143,144], a Holiday junction resolvase, two PCNAs, replication factor C, RNase H, and a type II DNA topoisomerase [145].

Initiation of virus DNA synthesis is followed by transcription of late genes. Currently very little is known about late transcription in the chloroviruses. We originally reported that late mRNAs might lack poly A tails [146]. These studies relied on pulse labeling infected cells with [^3^H]-adenine. However, we now know that PBCV-1 infection causes a rapid and sustained depolarization of the host membrane. This depolarization leads to inhibition of many secondary active transporters reliant on membrane potential and this changes solute uptake generally and suppresses the uptake of exogenous bases including adenine [118]. Therefore, our current understanding is that late viral mRNAs are polyadenylated.

### 5.5. Virus Assembly, Maturation and Release

At 3 to 4 h p.i., assembly of PBCV-1 capsomers begins in localized regions of the cytoplasm, which become prominent 4 to 5 h p.i. (Figure 1G). These localized regions, called virus assembly centers or virus factories, consist of host cisternae that are derived from the ER next to the nuclear membrane (Figure 1G) [48]. The cisternae are localized at the periphery of the viral assembly centers and are cleaved into single bi-layered membranes, which then move to the central region of the assembly centers. Capsomers form around these membranes leading to the formation of empty virions at the periphery of the virus assembly centers where DNA packaging occurs [47,48].

DNA packaging may involve a virus-encoded DNA packaging ATPase. However, the putative DNA packaging ATPase is part of a larger CDS (A392R), which consists of two predicted domains separated by a caspase cleavage site. We predict that virus DNA packaging begins after either a host- or virus-encoded enzyme with caspase-like activity cleaves A392R to produce a functional DNA packaging enzyme [147]. (Note: earlier studies established that a caspase-like cleavage activity in phycodnavirus Emiliana huxleyi virus (EhV) was involved in EhV replication [148,149]).

By 5 to 6 h p.i. infectious PBCV-1 progeny accumulate in the cytoplasm and localized lysis of the host cell membrane and cell wall releases progeny at 6 to 8 h p.i. (Figure 1I). Each infected algal cell releases up to ~1000 particles, of which ~25% form plaques. Mechanical disruption of cells releases infectious virus particles 30 to 50 min prior to cell lysis, indicating the virus is mature in the cell and that it does not acquire its glycoprotein capsid by budding through the host plasma membrane as it is released from the cell [35]. Currently nothing is known about the clock mechanism that must operate to time the release of the nascent virions; however, the viruses encode several cell wall degrading enzymes that presumably function for the virus to escape the cell. A schematic diagram of the predicted PBCV-1 replication cycle is reported in Figure 8.

A crude enzyme mixture can be prepared from viral lysates, called lysin, that has been used to prepare protoplasts of all four hosts of the chloroviruses [100]. Presumably lysin consists of a mixture of the chlorovirus encoded cell wall degrading enzymes mentioned above.

Most of the chloroviruses encode a functional Cu/Zn superoxide dismutase (SOD1) [151]. The gene for the PBCV-1 enzyme (*a245r*) is expressed as a late gene and the protein is packaged in the virion. Reactive oxygen was measured in *C. variabilis* NC64A cells infected with either PBCV-1 or another NC64A chlorovirus NY-2A, which lacks the *a245r* gene (members in this subclade are devoid of the SOD1 gene). Superoxide accumulated rapidly during infection by both viruses; however, the amount was lower in the PBCV-1 infected cells and we predict that the presence of the PBCV-1 encoded Cu/Zn superoxide dismutase contributed to the lower amount. Furthermore, the virus NY-2A had a lower burst size and it took two to three times longer to replicate than PBCV-1. We suspect that these two properties might be due, in part, to the lack of a Cu/Zn superoxide dismutase-encoding gene [151].

Adding to the complexity of pathogenesis, chloroviruses encode proteins that manipulate the cellular ubiquitin-proteasome proteolytic pathway. They encode several proteins analogous to the cellular Skp1-Cul1-F-box (SCF) ubiquitin ligase complex. PBCV-1 encodes two F-box-like proteins, ankyrin-repeat proteins (A682L and A607R) that function as binding partners for a virus-encoded Skp1 protein (A039L) [152]. These ankyrin-repeat proteins have a C-terminal Skp1 interactional motif that functions similarly to cellular F-box domains and can bind Skp1 proteins from widely divergent species. The chlorovirus ankyrin-repeat and Skp1 proteins interact exclusively with corresponding monophyly clusters amongst three chlorovirus-types, suggesting a partnership function tailored to the host alga. In addition to viral SCF-mimicry, 11 SAG viruses, including ATCV-1, and one NC64A virus, NY-2A, encode their own viral ubiquitin that may aid viral replication and evasion from cellular immune defenses. The viral-encoded ubiquitins have 97% amino acid identity with human ubiquitin.

PBCV-1 also encodes putative enzymes involved in protein degradation, including a ubiquitin C-terminal hydrolase (A105L) and RING finger E3 ubiquitin ligase (A481L). Taken together, this group of chlorovirus-encoded ubiquitin-related genes creates a collection of proteins that threaten the host’s ubiquitin-proteasome machinery. These viral-encoded proteins may engage the ubiquitin system through ubiquitin-binding motifs, another important specificity-providing component of the ubiquitin system. Finally, one would predict that host cytoskeleton elements must play a role in both the movement of the viral DNA to the nucleus and/or the formation of the virus assembly centers. This possibility was explored by adding several inhibitors of tubulin and actin activity to cultures during virus replication. However, none of the inhibitors had any effect on PBCV-1 morphogenesis even though in some experiments the inhibitors were added either 2 or 24 h prior to virus infection and at three times the concentrations that inhibited growth of uninfected algae [153]. These studies, however, pre-dated the understanding of virus-mediated membrane depolarization and the subsequent suppression of secondary transporters. These experiments should be re-examined with alternate approaches.

### 5.6. Regulation of Nucleotide Metabolism

PBCV-1 infection of *C. variabilis* NC64A results in the production of approximately 1000 particles per cell in a 6–8 h lytic cycle, of which 25%–30% form plaques. As noted above, virus amplification represents a 4-fold or more increase in DNA content of the cell with a shift in G+C content of 67% from the uninfected host cell to 39% for the virus. The pool of nucleotide triphosphates needed for the DNA increase comes in part from the salvage pathway of degraded host DNA, and in a larger part from de novo synthesis.

The DNA sequences for both the host cell [29] and PBCV-1 [43] have been determined and the protein coding genes annotated. PBCV-1 encodes eight proteins that contribute to pyrimidine biosynthesis, including aspartate transcarbamoylase, both large and small subunits of ribonucleotide reductase, thioredoxin, glutaredoxin, dCMP/dCTP deaminase (biDCD), cytosine deaminase, dUTP pyrophosphatase, and thymidylate synthase X [154]. All of these genes are expressed in the early phase of infection prior to the initiation of DNA synthesis.

Both the dCMP/dCTP deaminase activities are sensitive to dTTP as an inhibitor, and the dCMP deaminase activity is sensitive to dCTP as an activator. The viral aspartate transcarbamoylase gene encodes the catalytic enzyme, but not the regulatory elements [155]. Thus, expression of this gene renders the cell insensitive to the typical nucleotide triphosphate feedback mechanism characteristic of this important enzyme.

Annotation of the 8-fold coverage map of *C. variabilis* NC64A reveals the expected pathway for pyrimidine metabolism. The pathway includes the typical entry point of carbamoyl phosphate from the urea cycle condensing with l-glutamate from the glutamate metabolism pathway to form n-carbamoyl-l-aspartate via aspartate transcarbamoylase, a biosynthetic event that is typically highly regulated to achieve an appropriate balance of nucleotides for RNA and DNA synthesis.

Although annotation of the host genome supports a robust pyrimidine biosynthesis pathway, comparison with the virus indicates certain deficits that may bias synthesis to a relatively higher G+C content. These “deficits” are apparently complemented by the virus genes, when the cell is re-programmed for de novo synthesis of a relatively low G+C content [154]. Metabolomic analyses are needed to evaluate the trajectory of the pyrimidine biosynthesis pathway in this early and critical phase of the virus infection.

## 6. Chlorovirus Genes and Biotechnology

The issue of why the choroviruses encode DNA restriction/modification systems was a puzzle for a while. Bacteria are believed to encode these systems to keep out foreign DNA and viruses. However, in this case the viruses encode the enzymes and the restriction endonucleases are packaged in the virions but not the corresponding DNA methyltransferases. We now believe that the restriction endonucleases enter the cell along with the viral genome during infection and that they are responsible for the rapid degradation of the host chromatin during infection [124]. However, before the rapid degradation of the host chromatin was known we proposed that the restriction endonucleases might prevent the cell from being infected by more than one virus. Therefore, we simultaneously infected the host with two different types of viruses for two reasons: first, to determine if one function of the restriction endonucleases was to exclude other viruses, and second, to determine if the algal host could support simultaneous replication of two different viruses [156]. However, plaques, which were picked by replicate plating and identified by immunoblotting, that arose from single cells inoculated with two viruses only contained one of the two viruses about 90% of the time, i.e., the viruses exhibited mutual exclusion. However, the restriction endonucleases are not responsible for this exclusion [156].

Many chlorovirus encoded proteins are either the smallest or among the smallest proteins of their class. Despite their small sizes, the virus-encoded proteins typically have all of the catalytic properties of larger proteins. The small sizes and the fact that the virus-encoded recombinant proteins that are often “user friendly” has resulted in the biochemical and structural characterization of many chlorovirus-encoded proteins, some of which were mentioned above and have led to commercial products and/or biotechnological tools for some of them. Examples include: (i) one of the smallest eukaryotic ATP-dependent DNA ligases and there are over 20 research publications on the enzyme (e.g., Ho et al. [143]; Krzywkowski & Nilsson [157]). The PBCV-1 encoded DNA ligase is sold commercially because it efficiently catalyzes the ligation of adjacent, single-stranded DNA primers splinted by a complementary RNA strand, referred to as SplintR ligase [158]. Therefore, combining this enzyme with quantitative PCR allows one to determine the precise number of specific mRNA molecules in a preparation; (ii) The smallest type II DNA topoisomerase, which cleaves unmodified dsDNAs 30 to 50 times faster than the human type II DNA topoisomerase [159]. This faster activity may be due to the fact that the enzyme needs to interact with 6mA containing virus DNA [160]; (iii) A small prolyl-4-hydroxylase that converts Pro-containing peptides into hydroxy-proline-containing peptides in a sequence-specific fashion [161]. The gene is present in all of the chloroviruses. However, its role in virus replication is unknown, and attempts to detect hydroxyl-proline in purified virion proteins have been unsuccessful [162]; (iv) The smallest histone methyl-transferase (named vSET), which trimetylates Lys^27^ in histone 3 [122,163]. This specific methylation is implicated in gene silencing and as mentioned above is involved in inhibiting host transcription during virus infection; (v) One of the smallest proteins to form a functional K^+^ channel named Kcv. Whereas, K^+^ channels from prokaryotes and eukaryotes often consist of several hundred amino acids, PBCV-1 Kcv consists of 94 amino acids [164] and chlorovirus ATCV-1 Kcv consists of 82 amino acids [165]. Despite their small sizes, both viral proteins form functional channels that have many of the properties associated with larger K^+^ channels such as selectivity, gating and sensitivity to inhibitors [166,167,168,169]. Kcv proteins readily assemble into tetramers and move to the plasma membrane in a variety of cells [170,171]. Thus, all of the structural requirements for correct assembly of the channel as well as for the basic functional properties of a K^+^ channel exist in Kcv proteins [172]. The chlorovirus-encoded K^+^ channels are under intensive investigation and there are over 60 publications on them.

One very important property of the chlorovirus Kcvs is that their amino acid sequences can differ slightly among viruses and consequently they can have different physiological properties. For example, the Kcv protein was identified in 40 additional NC64A viruses and differences in 16 of the 94 amino acids were observed that produced six Kcv-like proteins with amino acid substitutions present in most of the functional domains of the protein [173]. Because the channel is apparently required for virus replication, the Kcv variants are all functional; however, they can have different physiological properties including altered current kinetics and inhibition by cesium. The amino acid changes, together with different properties in the channels, can be used to guide site-directed mutations to identify key amino acids that confer specific properties to Kcv [174].

Chimeric Kcvs have been constructed by linking PBCV-1 Kcv to a voltage-sensing domain from a phosphatase enzyme [175] or a photo sensory module (called BLINK1, [176]), which results in a K^+^ channel that responds to either voltage or blue/red light, respectively. The properties of BLINK1 have recently been improved and the channel protein is now referred to as BLINK2 [177]. The fact that one can go from a very simple channel to a complicated channel in a single genetic step, along with some other properties such as the fact that the *Kcv* gene did not come from the host alga, led to the hypothesis that K^+^ channels might have evolved from chloroviruses [178]. Support for this hypothesis was provided by a recent report describing the coupling of the 82 amino acid Kcv from chlorovirus ATCV-1 with a glutamate-binding-domain from a rat; the channel is functional and displays the ligand recognition characteristics of GluA1 and the K^+^ selectivity of Kcv_ATCV-1_ [179]. However, the subsequent finding of more diverse virus-encoded K^+^ channels has tended to question this hypothesis [180].

In addition to the aforementioned K^+^ channel Kcv, some of the chloroviruses encode other functional channel/transporter proteins [181] including an aquaglyceroporin [182], a Ca^2+^-transporting ATPase [183], and a HAK/KUP/KT-like K^+^ transporter [184].

Some other chlorovirus encoded enzymes include: (i) A pyrimidine-dimer specific DNA glycosylase, which is a homolog of a phage T4 DNA repair enzyme [38]. The gene encoding the enzyme is present in all the chloroviruses. The PBCV-1 enzyme, referred to as CV-PDG, possesses a higher catalytic efficiency and broader substrate specificity than the T4-PDG (e.g., McCullough et al. [185]; Jaruga et al. [186]). The enzyme, which is not packaged in the virion but is expressed early, can repair UV-induced DNA damage in infected cells [38]; (ii) An aspartate transcarbamylase [155], as mentioned above this enzyme is important in regulating pyrimidine biosynthesis. However, the gene encoding this enzyme is only present in NC64A chloroviruses that have genomes of 40% G+C; (iii) A thymidylate synthase X [187], rather than the catalytically more efficient thymidine synthase A enzyme, which is present in most organisms; (iv) PBCV-1 encodes as many as seven predicted Ser/Thr protein kinases, some of which are packaged in the virus [45,188], and also a dual specific phosphatase. However, nothing is known about the role these kinases and the phosphatase play in the PBCV-1 life cycle; (v) All of the chloroviruses encode a member of the ERV1/ALR protein family that is involved in disulfide bond formation as well as a protein disulfide isomerase.

Almost all of the chloroviruses encode a homolog of the translational elongation factor-3, which is present in many fungi [189]. The yeast protein exhibits ribosome-dependent ATPase and GTPase activities that are not intrinsic to the fungal ribosome but are nevertheless essential for translation elongation in vivo. A BLAST search with the viral-encoded protein indicates that the gene is also present in many algae. The function of the protein, which is formed early in chlorovirus replication [190], is unknown.

Presumably, polyamines play an important role in the chlorovirus life cycle as PBCV-1 is the first virus to encode five functional enzymes involved in polyamine biosynthesis and catabolism including: ornithine decarboxylase (ODC) [191,192], agmatine iminohydrolase [193], N-carbamoylputrescine amidohydrolase [193], homospermidine synthase [194], and a polyamine acetyltransferase [195]. Not only is the PBCV-1-encoded ODC among the smallest known ODCs, the PBCV-1 enzyme is also interesting because it decarboxylates arginine more efficiently than ornithine [192]. Homospermidine synthase is a virion-associated protein. These five polyamine metabolic enzymes are highly conserved among all of the chloroviruses and so one would predict that they must serve an important role in chlorovirus biology and/or replication. Polyamines have been reported to be present in high concentrations in some viruses where they can neutralize up to 50% of the viral genome (e.g., Cohen [196]). However, this is unlikely to be their role in the chloroviruses because the polyamine concentrations, including putrescine, cadaverine, spermidine, and homospermidine, are so low that they can only neutralize ~0.2% of the virus phosphate residues in highly purified PBCV-1 particles [194]. Basic proteins and divalent cations apparently play a big role in neutralizing the PBCV-1 genome [120]. As reported in the ecology section below, the chloroviruses can attract *Paramecium bursaria*, which harbor the zoochlorellae hosts for the viruses, with a chemo-attractive agent [197] and maybe the agent is one or more of the polyamines.

## 7. Carbohydrate Manipulating Genes in Chlorovirus Biology

Chloroviruses are also unusual because they encode enzymes involved in sugar metabolism [198,199,200]. Three PBCV-1-encoded enzymes, UDP-glucose dehydrogenase [201], glutamine:fructose-6-phosphate amino transferase [201], and hyaluronan synthase (*has* gene) [202] contribute to the synthesis of hyaluronan (hyaluronic acid), a linear polysaccharide composed of alternating β-1,4-glucuronic acid and β-1,3-*N*-acetylglucosamine residues. Hyaluronan is part of the extracellular matrix of vertebrates and it is also present on the exterior surface of some pathogenic bacteria as a way to avoid the immune system [203]. All three PBCV-1 genes are transcribed early in virus infection and a dense fibrous hyaluronan network accumulates on the surface of the infected cells (Figure 9B) [204].

Hyaluronan is synthesized at the plasma membrane of eukaryotic cells; the alternating sugars are added and the newly synthesized hyaluronan is pushed out into the extracellular matrix of the cells [203]. Presumably the same thing happens in the chlorovirus-infected cells. However, here the newly synthesized hyaluronan has to pass through the algal cell wall, which is a network of polysaccharides. An analogy might be pushing a thread through a furnace filter. In fact, when the PBCV-1 *has* gene was expressed in a plant system, the hyaluronan bunched up between the plasma membrane and the cell wall [205]. Could the hyaluronan formed in the chlorella cells have a pilot protein attached?

The *has* gene is present in about 30% of the chloroviruses. Surprisingly, many chloroviruses that lack a *has* gene have a gene encoding a functional chitin synthase (*chs*) gene. Furthermore, cells infected with these viruses produce chitin fibers on their external surface [206]. Chitin, an insoluble linear homopolymer of β-1,4-linked *N*-acetyl-glucosamine residues, is a common component of insect exoskeletons, shells of crustaceans, and fungal cell walls.

A few chloroviruses encode both *chs* and *has* genes and form both chitin and hyaluronan on the surface of their infected cells [206,207]. Finally, some chloroviruses lack both genes and cells infected with these viruses produce no known extracellular polysaccharides [204]. The fact that many chloroviruses encode enzymes involved in extracellular polysaccharide biosynthesis, which require huge amounts of ATP for their synthesis, suggests that the polysaccharides are important in the viral life cycles. However, they do not appear to be required for chlorovirus replication in the laboratory. Furthermore, because all of the chloroviruses, including ones that lack the *has* and *chs* genes, have been isolated from natural sources in recent years, the genes do not appear to be essential for replication in native environments.

Two PBCV-1 encoded enzymes, GDP-d-mannose dehydratase (GMD) and GDP-4-keto-6-deoxy-d-mannose epimerase/reductase (GMER, also referred to as fucose synthase), comprise a three step pathway that converts GDP-d-mannose to GDP-l-fucose [208]. The PBCV-1 GMD is unusual because it can also convert the GMD product into GDP-d-rhamnose in the presence of NADPH. (Note: the chlorovirus ATCV-1 GMD homolog lacks the GDP-rhamnose forming activity [209].) Both fucose and rhamnose are in the glycans attached to the PBCV-1 MCP (Figure 3). Genes encoding the two enzymes involved in synthesizing fucose are present in all the NC64A and SAG viruses, but are absent in all the Pbi viruses. However, only a few of the chlorovirus GMDs can synthesize rhamnose in addition to fucose [209]. Other chloroviruses encode additional sugar metabolizing enzymes including a UDP-d-glucose-4,6-dehydratase [210] and a putative mannose-6-phosphate isomerase.

Typically, viruses use host-encoded glycosyltransferases and glycosidases located in the ER and Golgi apparatus to add and remove N-linked sugar residues from virus glycoproteins either co-translationally or shortly after translation of the protein. The virus glycoproteins are then transported to the host plasma-membrane where progeny viruses bud though the membrane and they only become infectious as they leave the cell [211]. However, unlike the process described above, glycosylation of the chlorovirus MCPs differs from that scenario because the viruses encode most, if not all, of the machinery for the process. Furthermore, the glycosylation process occurs in the cytoplasm [199,200].

The conclusion that the chlorovirus PBCV-1 MCP (protein Vp54, gene *a430l*) is glycosylated by a different mechanism than that used by other characterized viruses began from antibody studies in 1993 [212]. Rabbit polyclonal antiserum [213] particles inhibited virus plaque formation by agglutinating the virions. Spontaneously-derived antiserum-resistant, plaque-forming variants of PBCV-1 occur at a frequency of 10^−5^ to 10^−6^. At the time of the 1993 publication, four serologically-distinct classes were identified plus wild type virus; subsequently two additional antigenic variants have been isolated for a total of six antigenic variants. Polyclonal antisera prepared against members of each of these antigenic classes react predominately with the Vp54 protein equivalents from the viruses in the class used for the immunization. Each of the Vp54 proteins from the antigenic variants migrated faster on SDS-PAGE than those from the strains from which they were derived, indicating a lower molecular weight. However, all of the de-glycosylated MCPs migrated at the same rate on SDS-PAGE, indicating that the size differences were due to the attached glycans. In addition, the *a430l* nucleotide sequence in each of the antigenic variants was identical to the wild-type gene; these results verified that the polypeptide portion of Vp54 was not altered in the antigenic variants. Western blot analyses of Vp54 proteins before and after removing the glycans with trifluoromethane-sulfonic acid or altering the glycan with periodic acid, also supported the notion that the antigenic variants reflected differences in the Vp54 glycans, not the Vp54 polypeptide [212].

Subsequently, all experimental results indicate that Vp54 glycosylation occurs in the cytoplasm and independent of the ER and Golgi apparatus [212,213]. For example, unlike viruses that acquire their surface glycoprotein(s) by budding through the plasma membrane and only become infectious during their release from the host, intact infectious PBCV-1 particles accumulate inside their host 30–40 min prior to virus release [35]. The identification of the glycan-linked Asn residues in the PBCV-1 Vp54 by X-ray crystallography provides additional evidence that Vp54 glycosylation does not involve host ER glycosyltransferases [50] because none of the Vp54 glycan-linked Asn residues reside in an Asn-X-Thr/Ser sequon sequence commonly recognized by ER glycosyltransferases [214]. Finally, the PBCV-1 Vp54, like the PBCV-1 encoded glycosyltransferases, lacks an ER signal peptide. The unusual nature of the virus-encoded glycosylation pathway led us to suggest that this glycan synthesis pathway might pre-date the evolution of the Golgi-ER glycosylation pathway [53,204]. The recent report that the evolutionary ancestor of giant viruses, including the chloroviruses, predated the origin of modern eukaryotes [215] is consistent with this suggestion.

PBCV-1 encodes at least eight putative glycosyltransferases (GTs), some of which, if not all, participate in glycosylating the virus Vp54 protein [199,213,216,217]. Ongoing experiments have identified the role of three PBCV-1-encoded GTs with four GT activities that are involved in the synthesis of the glycan (Figure 10A). Two of the three GTs were previously annotated as GTs but the third protein was only identified in a recent study [213].

Most significantly, the PBCV-1 gene *a064r* encodes a protein with three domains: domain 1 has a β-l-rhamnosyltransferase activity [213,216,217,218], domain 2 has an α-l-rhamnosyltransferase activity [213,217], and domain 3 encodes a methyltransferase that methylates the 2 position in the terminal α-l-rhamnose unit (Figure 10B) [217]. In retrospect the *a064r* gene is interesting because in the late 1990s we tried to make recombinant viruses between different PBCV-1 anti-genetic variants to create wild-type viruses [216]. We were successful with some combinations and not with others. It turns out that the unsuccessful crosses were with variants that were due to changes in the three domains of the *a064r* gene, which would lead to rare recombinants.

The *a075l* gene is predicted to encode a β-xylosyltransferase that attaches the distal d-xylose unit to the l-fucose that is part of the n-glycan core region. Gene *a071r* is predicted to encode a GT that is involved in the attachment of a semiconserved element, α-d-rhamnose to the l-fucose in the core region. The *a111/114r* gene is predicted to encode a protein with three domains. The N-terminal domain is a xylosyltransferase, the middle domain is a galactosyltransferase and the C-terminal domain is a fucosyltransferase [217].

Finally, we predict that gene *a301l* encodes the protein that links glucose to the Asn [217]. Ongoing experiments include cloning and expressing the predicted GTs and assaying them for the predicted biochemical activity. Another set of experiments are addressing the question: are the glycan precursors attached to a lipid carrier such as undecaprenol-phosphate, which serves as an intermediate in bacterial peptidoglycan synthesis [219], or dolichol diphosphate, which serves the same function in eukaryotic cells [214], or are the sugars added sequentially to the glycan intermediates attached to the protein?

## 8. An Unknown but Interesting CDS

One additional PBCV-1 encoded CDS (A312L) should be mentioned because the gene is one of the most highly expressed genes [75] and its 33 kDa protein is made in large amounts during PBCV-1 replication [49]. The gene is expressed both early and late in the virus replication cycle and the protein, which is coded by all of the chloroviruses, is not packaged in the virion. The 5′-UTR region of the gene can function as a translational enhancer in *Arabitopsis* [220]. The function of this protein is completely unknown but it does have close homologs in the Ostreococcus viruses.

## 9. Natural History of the Chloroviruses

Short [221] wrote an excellent review on the ecology of viruses that infect eukaryotic algae, including the chloroviruses. Chloroviruses are ubiquitous in inland water habitats throughout the world (e.g., Van Etten et al. [80]; Zhang et al. [222]; Yamada et al. [223]). Initially, NC64A viruses were only found in North America and Pbi viruses were only found in European waters. However, subsequent experiments found Pbi viruses in water samples collected from the Northern United States and Canada, as well as higher elevations in the United States. Likewise, NC64A viruses were present in waters from Southern Europe, the Middle East, and China and Japan. Both NC64A and Pbi viruses have been found in water collected in Australia and in South America. The localized regions where the viruses occur presumably reflect the growth of the protists that have the host zoochlorellae, but currently there are no known ecoregional boundaries.

The chloroviruses are sometimes quite rare, but at other times, they show major spikes in abundance (up to 10^5^ PFU per mL). For example, in a three year study of an urban lake in Lincoln, NE three chlorovirus types (NC64A, SAG, and Osy) fluctuated in abundance throughout the year, with a peak during the late spring and a second one during late fall [13]. (No Pbi viruses were detected in this study). Even in low-viral-abundance months, infectious chlorovirus populations were maintained, suggesting either that the viruses are very stable [224] or that there is ongoing viral production in natural hosts. It is unknown if chloroviruses replicate exclusively in zoochlorellae symbiotic with paramecia and heliozoae or if the viruses have another host(s). However, attempts in our laboratory to find other hosts have been unsuccessful to date [225]. Also it is unknown if these zoochlorellae exist free of their hosts in natural environments because replication of the zoochlorellae in native water (free of infectious chloroviruses) is very slow, if it occurs at all [226].

An important question has been: how do chloroviruses infect zoochlorellae to produce the occasionally high virus titers observed in nature? It is known that chloroviruses can attach to the external surfaces of paramecia in the ciliary pits of the cortex and in the buccal cavity regions without actually infecting the paramecia [17,227]; virus attachment to the paramecia does not appear to involve the spike structure (Figure 11). Therefore, the viruses are in good position to infect zoochlorellae if the paramecium cell, which can harbor 300–600 zoochlorellae [16], is ruptured either artificially or naturally by an aquatic predator. In fact, artificial disruption of paramecia, by either sonication or neutral detergents, leads to an increase in virus titers of ~10^2^ PFUs per zoochlorella 8 to 12 h later [228]. Interestingly, the chloroviruses do not attach to a symbiont-free species of *Paramecium caudatum,* which likely lacks the virus specific receptor [227].

Several types of aquatic predators, including protists, copepods, nematodes, and planarians, consume *P. bursaria* (e.g., DeLong [229]). Therefore, some of these predators were examined for their effect on the replication of chloroviruses in microcosm experiments. Foraging by the cyclopoid copepod (*Eucyclops agilis*) disrupts the physical barrier between chloroviruses and their host zoochlorellae inside the paramecium by engulfing the entire paramecium. Subsequence passage of the paramecia through the copepod digestive system results in the release of fecal pellets containing viable zoochlorellae and chloroviruses (Figure 12). Incubation of these isolated fecal pellets leads to rapid virus amplification [228].

Another aquatic predator, *Didinium nasutum*, especially the smaller ones, that feed on *P. bursaria* can rupture the protist by tearing it apart, which also releases the zoochlorellae [230]. This process, which is referred to as “messy feeding”, also leads to rapid chlorovirus amplification. The ciliate *Bursaria truncatella* can also disrupt *P. bursaria* leading to an increase in chlorovirus titer [231]. These findings give rise to the concept of predation being an “ecological catalyst”.

The systems described above depend on the chloroviruses being attached to the *P. bursaria* surface without actually infecting the paramecia. Currently, nothing is known about the nature of this attachment process. The accumulation of infectious virions on the cell surface, which can number in the hundreds [228], could occur through random contacts between *P. bursaria* and chlorovirus particles as the paramecia move through the water. However, ongoing experiments indicate that the chloroviruses release a chemotactic agent that attracts *P. bursaria* [197]. To our knowledge this is the first example of a virus attracting its host through long-distance chemical signaling although in this example the virus is attracting the organism that is carrying the host. Currently, experiments are evaluating the chemical nature of this chemo-attractant.

## 10. Resistance to Chlorovirus Infections

One report explored the possibility that the chlorovirus host alga, *C. variabilis* NC64A, might encode and might express the same genes that higher plants use to respond to virus infection, including genes involved in RNA silencing [232]. The study identified 375 plant homologs that were coded by *C. variabilis* NC64A. RNA-Seq data showed that 325 of these 375 homologs were expressed in uninfected and early PBCV-1 infected (i.e., less than 60 min p.i.) cells. For each of the RNA silencing genes to which homologs were found, mRNA transcripts were detected in uninfected and infected cells. Therefore, *C. variabilis* NC64A, like higher plants, may employ certain RNA silencing pathways to defend itself against virus infection. To our knowledge this was the first examination of RNA silencing genes in algae beyond core proteins, and the first analysis of their transcription during virus infection.

It is easy to isolate *C. variabilis* NC64A cells that develop resistance to infection by PBCV-1. With the exception described below, resistance is due to the inability of the viruses to attach to the alga. That is, the host receptor, which is predicted to be a carbohydrate [101], is altered.

The exception to this concept was the finding that some Osy chloroviruses, e.g., virus Osy-NE5, that infect *C. variabilis* Syngen 2–3, can also attach to *C. variabilis* NC64A cells and initiate infection. Essentially no infectious viruses are produced from this interaction [94] and ongoing studies are evaluating the point at which the Osy-NE5 replication cycle is blocked.

Currently there is no evidence to indicate that the chloroviruses have a lysogenic phase. However, the chloroviruses can form a carrier-state relationship (also called pseudo-lysogeny), where at any one time a small population of algae are continually infected by virus. This type of relationship occurs with bacteriophage (e.g., see Hayes [233]). On two occasions we have observed a carrier state relationship between PBCV-1 and *C. variabilis* NC64A in the laboratory. That is, algal colonies occasionally grow in PBCV-1 plaques. Colonies picked from these plaques continually produced low concentrations of PBCV-1 (~10^5^ PFU/mL), even after repeated sub-culturing over several months. PBCV-1 did not hybridize to DNA isolated from single-colony isolates of *C. variabilis* NC64A that were isolated in the presence of PBCV-1 antibody, which eliminates PBCV-1. This experiment rules out lysogeny. However, about 25% of these isolates, freed of PBCV-1, could re-establish the carrier-state relationship when re-inoculated with PBCV-1. The remaining cultures were either sensitive or resistant to PBCV-1 [234].

## 11. Evolutionary History

A phylogenetic analysis of 32 concatenated conserved genes from 41 chloroviruses that infect three zoochlorellae hosts, i.e., 14 NC64A viruses, 14 Pbi viruses, and 13 SAG viruses, revealed three important features about chlorovirus evolution (Figure 13A) [93]: (i) Even though the 41 chloroviruses were isolated from diverse locations across five continents, the phylogenetic trees show that viruses infecting the same zoochlorella host clustered in monophyletic clades. This observation suggests that the most recent common ancestor of each chlorovirus subgenus already infected the same algal host lineage as today’s representatives and that the evolutionary events that led the viruses to adapt and specialize on a given host occurred only once in their history; (ii) The branching pattern of the three main virus clades does not match the phylogeny of the 18S RNA alignment of their zoochlorellae hosts (Figure 13B), which eliminates the simplest co-evolution scenario whereby the algae and virus lineages separated at the same time. Instead, the phylogenetic evidence suggests that the chloroviruses have changed hosts at least one time in their evolutionary history; (iii) While most of the newly sequenced chloroviruses are close relatives of previously sequenced chloroviruses, the basal and isolated phylogenetic position of virus NE-JV-1 within the Pbi virus clade is the first representative of a previously unknown subgroup of chloroviruses. NE-JV-1 only shares 74% amino acid identity on average with the other Pbi viruses in the 32 core proteins used for the phylogeny reconstruction. For comparison, the within-clade average protein sequence identity was 93%, 95%, and 97% identity for NC64A, SAG, and Pbi (excluding NE-JV-1) viruses, respectively. Between clades, the protein sequence identity ranged from 63% (NC64A vs. Pbi viruses), to 71% (Pbi vs. SAG viruses). The most recently discovered chlorovirus group, the Osy viruses, are interesting because phylogenetic analysis of 29 of the 32 conserved concatenated core proteins of chlorovirus Osy-NE5 indicates that it resides between two separate phylogenetic subclades of NC64A viruses (Figure 13A) [94].

The evolutionary origin of individual chlorovirus genes is also interesting. Phylogenetic analysis identified seven protein families where the viral protein clades branched next to *C. variabilis* NC64A homologs, reflecting potential horizontal gene transfer (HGT) between the virus and their host. For two of the seven genes, the likely direction of the HGT was from the current host to the chlorovirus ancestor. For four genes, *C. variabilis* NC64A is the only plant organism to have these virus genes; thus, their vertical inheritance from an ancestor is more unlikely and suggests that the virus genes were transferred to the host zoochlorellae [93]. Thus, although large viruses are often described as mainly evolving by recruiting genes from their hosts (e.g., Koonin & Yutin [235]), this conjecture does not appear to hold true for the chloroviruses and their current hosts.

Oligonucleotide frequency analysis led to one of the more surprising discoveries in the Jeanniard et al. [93] study; i.e., 12% to 18% of the chlorovirus predicted protein families are encoded by genes of recent extrinsic origin (i.e., after chlorovirus divergence)—most of which are ORFans. However, clues as to their potential donor genomes are lacking. The occurrence of these putative genes supports the concept that viruses might be the source of new proteins (e.g., Filee & Forterre [236]; Koonin & Dolja [237]).

As noted above, some chlorovirus genes have introns. In fact, virus PBCV-1 was the first virus reported to encode three types of introns: (i) a self-splicing group IB intron in the TFIIS transcription factor encoding gene *a125l* [238], (ii) a nuclear-located, spliceosomal processed intron in the DNA polymerase gene *a185r* [142], and (iii) a small intron in a tRNA gene [239]. Self-splicing introns have also been identified in other chlorovirus genes [239]. An interesting observation from an evolution standpoint deals with the fact that some of the pyrimidine dimer-specific glycosylase (*cv-pdg*) encoding genes, which are present in all of the chloroviruses, have a splicesomal-processed intron.

The *cv-pdg* gene was cloned from 42 NC64A chloroviruses that were isolated over a 12-year period from diverse geographic regions and there was no obvious geographic correlation between *cv-pdg* intron-containing and intron-lacking viruses [240]. The *cv-pdg* gene from 15 of these 42 viruses contains a 98-nucleotide intron that is 100% conserved among the viruses and another four viruses contain a truncated 81-nucleotide version of the 98-nucleotide intron in the same position that is nearly 100% identical (one virus differed by one nucleotide). In contrast, the nucleotides in the *cv-pdg* exons from the intron-containing viruses were 84% to 100% identical. The 100% identity of the 98-nucleotide intron sequence in the 15 viruses and the near-perfect identity of the 81-nucleotide sequence in another four viruses imply strong selective pressure to maintain the DNA sequence of the intron when it is in the *cv-pdg* gene. However, the ability of intron-plus and intron-minus viruses to repair UV-damaged DNA in the dark was nearly identical [240]. These findings contradict the widely accepted dogma that intron sequences are more variable than exon sequences. Another example of the highly conserved type I intron in the *cv-pdg* gene was its identical presence in two chloroviruses isolated ten years apart and at locations thousands of miles apart [240].

For comparative purposes, we also determined the sequence and conservation of a similar type of splicesomal-processed intron that is present in the PBCV-1 DNA polymerase gene (*dnapol*) [142], and its flanking regions in the same 42 NC64A viruses mentioned above [241]. [Note, the DNA polymerase encoding genes from the Pbi and SAG viruses lack the intron.] Thirty-eight of the 42 NC64A viruses contained a 101-nucleotide intron and the remaining four had an 86-nucleotide intron located in the same position in *dnapol*. However, unlike the intron in the *pdg* gene, the intron sequence in the *dnapol* gene was conserved (83% to 100% identity) to about the same extent as the coding regions of the gene (78% to 100% identical).

The chloroviruses, especially PBCV-1 and its host *C. variabilis* NC64A, are serving as a model system to address important subjects like: the repeatability of evolution, e.g., parallel evolution at the phenotypic and genotypic levels (e.g., Frickel et al. [242]). Co-evolution experiments demonstrate that selection and population sizes are changing and affect each other, and this parallelism is apparent in both the virus and host, where the population size appears to be a driver of the rapid evolution. However, the cost of this parallel evolution may result in a loss of intraspecies diversity, shaping community structure and functions. When ecological pressure on the algal host is added (predation), the rates of evolution are dampened [242]. These types of studies indicate the utility, flexibility and robust nature of the chlorovirus system for evaluation of extended concepts of ecology and evolution, beyond virological orthodoxy.

In the broader evolution picture, phylogenetic analyses of numerous chlorovirus genes indicate that the viruses most closely related to the chloroviruses are members of the prasinoviruses (also in the family *Phycodnaviridae*). This genus includes viruses that infect the smallest eukaryotic cell *Ostreococcus*, and related species in the class *Mamiellophyceae* (e.g., Derelle et al. [243]). However, it is interesting that the chloroviruses evolved mainly by gene duplications and losses of genes belonging to large paralogous families (including movements of diverse mobile genetic elements), whereas Micromonas and Ostreococcus phycodnaviruses derive most of their genetic novelties though HGTs [244,245].

Viruses in the family *Phycodnaviridae*, including the chloroviruses, are proposed to have an ancient, common evolutionary ancestry with some other large dsDNA viruses in the *Poxviridae*, *Iridoviridae*, *Ascoviridae*, *Asfarviridae*, *Mimiviridae, Marseillivirdae,* and *Pithoviridae* virus families. However, many more large DNA viruses are rapidly being discovered including Pandoraviruses, Faustoviruses, Mollivirus, Kaumoebavirus, Cetratvirus, Pacmanvirus, and Orpheovirus (e.g., Colson et al. [246]) and the evolutionary relationships among these viruses is just starting to be analyzed [235,247,248]. Collectively, these giant viruses are referred to as nucleocytoplasmic large dsDNA viruses (NCLDVs) [235,249,250,251] and it has been proposed that all these viruses should be included in a new order named *Megavirales* [252]. Using five genes that are nearly universal in the NCLDVs, a phylogenetic tree indicates that the NCLDVs form three major branches, with the *Phycodnaviridae* forming one branch with the *Mimiviridae* and the Pandoraviruses [235,247].

Although a common evolutionary ancestry of NCLDVs is generally accepted for at least some of the viruses considered to be NCLDVs, there are lively discussions on the role these viruses played in the evolution of eukaryotes. These discussions, which are likely to continue for some time, include the following questions: (i) was the evolutionary ancestor of the NCLDVs a free-living organism at one time that became trapped in other organisms so that they have been losing genes with time (e.g., Claverie & Albergel [253]) or did they arise from smaller viruses (e.g., Yutin et al. [254]; Koonin & Yutin [247])? (ii) Did the NCLDV genes arise from the original gene pool that led to prokaryotes and eukaryotes, or did they obtain many of their genes from smaller viruses (e.g., Legendre et al. [255]; Nasir et al. [256]; Colson et al. [246]; Koonin & Yutin [235])? (iii) Are the algal virus DNA polymerases near the root of the clade containing all eukaryotic DNA polymerase delta members [257]? (iv) Should the NCLDVs be included in the tree of life? See comments by seven groups in Nat. Rev. Microbiol. [258]. (v) Did primitive NCLDVs give rise to the eukaryotic nucleus or vice versa [259,260]? (vi) Does the structure of the chlorovirus MCP, which resembles MCPs from smaller dsDNA viruses with hosts in all three domains of life (human adenovirus, bacteriophage PRD1, and Archaea virus STIV), mean that these three viruses have a common evolutionary ancestor with the NCLDVs, despite the lack of amino acid sequence similarity among their MCPs (e.g., Krupovic & Bamford [60]; Holmes [261])?

## 12. Chloroviruses in Mammalian Biology

The question as to whether the choroviruses play a role in human and animal health arose several years ago when Dr. Robert Yolken of Johns Hopkins University School of Medicine (JHU) contacted us because they had discovered chlorovirus-like RNA sequences in brain tissues from individuals who had been afflicted with serious mental illnesses, such as schizophrenia and bipolar disorder. This was the first suggestion that expression of chlorovirus genes might have consequences on human health. Subsequent JHU research studies revealed that individuals were often seropositive for chlorovirus ATCV-1 proteins, suggesting an immunologically significant chronic exposure to ATCV-1 [262].

These unexpected results prompted additional experiments. Metagenomic sequencing of throat swab samples obtained from 92 individuals without a psychiatric disorder or serious physical illness, who were participating in a study that included measures of cognitive functioning, revealed that one or more ATCV-1 DNA fragments were present in 40 (43.5%) of the 92 subjects [263]. The presence of ATCV-1 DNA was not associated with demographic variables but was associated with a modest but statistically significant decrease in their performance on cognitive assessments of visual processing and visual motor speed.

The effects of ATCV-1 on mice were also explored. Introduction of ATCV-1-infected *C. heliozoae* into the intestinal tract of 9- to 11-week-old mice by gavage resulted in a subsequent decrease in performance in several cognitive traits including ones involving recognition memory and sensory-motor gating. No changes were observed in mice gavaged with non-infected *C. heliozoae* cells. The cognitive tests were conducted six or more weeks after the one-time gavage event. The mice were then sacrificed 26 weeks after gavaging with ATCV-1-infected algae and ~1300 of the mouse genes in the hippocampus exhibited altered expression. These genes comprised pathways related to synaptic plastidity, learning, memory formation, and the immune response to viral exposure [263].

Infectious ATCV-1 (measured by the plaque assay) was recovered from the spleen of mice at four weeks after the original gavaging of the mice [262]. In a second mouse experiment ATCV-1 was injected intra-cranially one time and slightly different procedures were used to monitor mouse behavior; this experiment also resulted in statistically significant cognitive changes in the virus-challenged mice [264].

Infection of either murine RAW264.7 cells or mouse primary macrophages resulted in a small amount of ATCV-1 replication, whereas chlorovirus PBCV-1 was not replicated [265]. Moreover, starting at 24 h post-challenge, RAW264.7 cells exhibited cytopathic effects, annexin V staining, and cleaved caspase 3. Activation of ERK MAP kinases occurred in these cells by 30 min post-challenge, which preceded the expression of interleukin-6 and nitric oxide. We hypothesize that ATCV-1 persistence in and induction of inflammatory factors and transit of these macrophages into the brain is a mechanism by which chloroviruses might alter cognitive performance. This hypothesis is based on the well-established theory that neuroinflammation impairs cognition and behavior [266,267,268]. While otherwise healthy individuals could have cognitive and behavioral impairments that are undetected, many neurological diseases are brought about or exacerbated by excessive inflammation. These diseases include multiple sclerosis, amyotrophic lateral sclerosis (ALS, also known as Lou Gehrig’s disease), Alzheimer’s, and Parkinson’s diseases. Therefore, it is possible that ATCV-1 persistence in macrophages with inflammatory cytokine could impact these neuroinflammatory diseases.

Since the initial studies with the JHU research group, several major studies have reported that chlorovirus genomes can be part of the human virome (e.g., Broecker et al. [269,270]; Liu et al. [271]; Moustafa et al. [272]), meaning that exposure to and persistence of chloroviruses in humans is apparently pervasive worldwide.

More recently, we explored the possibility that chloroviruses might be associated with motor neuron diseases (MND). MND is seen in pathologies ranging from a rare manifestation called ALS to a universal manifestation called human aging. ALS is a type of MND that has familial (genetic) causes and sporadic unknown causes. We discovered that serum from a cohort of patients with ALS and healthy control subjects have antibodies to chlorovirus ATCV-1, confirming either significant prior exposure to this virus, or to a similar antigen, in the general population. Moreover, we found that one type of antibody, IgG1, was significantly higher in ALS patients than healthy controls, suggesting that ATCV-1 might contribute to ALS and/or MND in general [273]. We then set out to test this hypothesis, which was prompted by the finding that ATCV-1 encodes a functional Cu/Zn superoxide dismutase (SOD) [151] and one type of familial ALS is caused by a polymorphism in human SOD1. Therefore, we adopted an ALS animal model, SOD1G93A-transgenic mice. These mice are genetically modified to express the mutant form of human SOD1 that is associated with familial human ALS and these mice develop symptoms of ALS-like MND by 150 days of life and die by 165 days. Intracranial infection with ATCV-1 significantly accelerated the progression of ALS-like symptoms in the SOD1-transgenic mice [273].

To summarize, we cannot say definitively that at least certain chloroviruses play a role in some of these disease situations. However, it is clear that many humans have either been exposed to chlorovirus ATCV-1 or at least some other protein that resembles the virus antigen(s). It is also clear that ATCV-1 DNA fragments and, in some cases, transcription products exist in some human viromes. The results are interesting enough that they merit further investigation.

## 13. Perspectives

For the past 40 years, the chloroviruses have provided many interesting and unexpected findings and concepts to the scientific community. This is reflected in that currently there are over 450 publications on the chloroviruses and their gene products. However, it is also clear that these studies are only scratching the surface on understanding these ubiquitous viruses. Chloroviruses are clearly playing roles in many aspects of biology, including symbiosis, virus ecology and maybe even human health. In addition, the chloroviruses code for many interesting proteins including some that have potential for commercial exploitation. However, it is also true that about half of the putative proteins coded by these viruses have unknown functions and they often have no homology to proteins in the public databases except with other chloroviruses. Who knows what more unexpected chlorovirus-encoded proteins await discovery? It should be noted that the chloroviruses have not only provided some biological surprises, they have also occasionally served as models for several physical measurements (e.g., Yonker et al. [274]; Sirotkin et al. [275]; Lee et al. [276]; Pande et al. [277]).

Genetics should be exploited to study the chloroviruses more than it has been. For example, Tessman [278] reported that PBCV-1 temperature-sensitive mutants could be selected and he used them to demonstrate genetic recombination at frequencies of about 1%. Subsequently, the only other report using genetics to study the viruses was with the PBCV-1 anti-genetic variants [216]. Finally, the development of efficient systems that allow researchers to conduct molecular manipulations of the chlorovirus genomes would be a huge boost to the study of chloroviruses and the phycodnaviruses in general. Such techniques, like the CRISPR system, are currently lacking, although we continue to try and develop such systems.

## Figures and Tables

**Figure 1 viruses-12-00020-f001:**
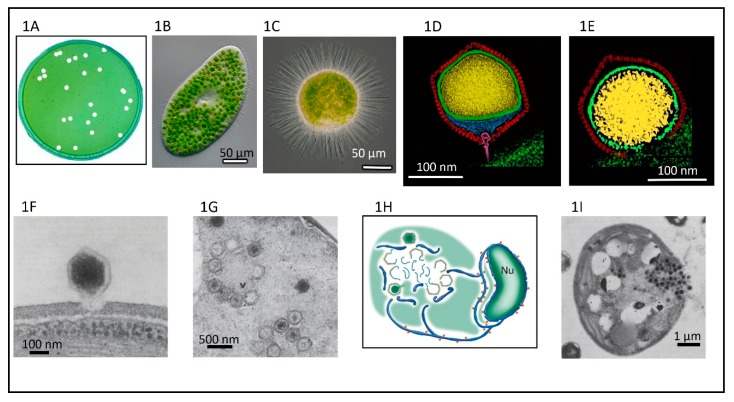
Chlorella cells and chlorovirus Paramecium bursaria chlorella virus-1 (PBCV-1). (**A**) Plaques formed by PBCV-1 on a lawn of *C. variabilis* NC64A. (**B**) The ciliate *Paramecium bursaria* and its symbiotic chlorella cells. (**C**) The heliozoon *Acanthocystis turfacea* and its symbiotic chlorella cells. (**D**) Cross section of a five-fold averaged cryo-EM image of PBCV-1 reveals a long narrow cylindrical spike structure at one vertex and the viral internal membrane (green) surrounding the viral genome asymmetrically. (**E**) Cross-section of a five-fold averaged cryo-EM of PBCV-1 as the virus is getting ready to release its DNA into the host cell. (**F**) Attachment of PBCV-1 to the algal cell wall and degradation of the wall at the point of attachment. This occurs within 1–3 min post infection (p.i.). (**G**) PBCV-1 particles assemble in defined areas in the cytoplasm named virus assembly centers at ~5 h p.i. Note both DNA containing (dark centers) and empty capsids. (**H**) A model showing that the origin of the PBCV-1 internal membrane arises from nuclei-derived cisternae, which serve as precursors for the single bi-layered virus membrane. Note, the membrane serves as the template for the capsid structures to form virus particles. (**I**) Localized lysis of the cell plasma membrane and cell wall and release of progeny viruses at ~7 h p.i. Panels **D** and **E** are from the cover of J. Virology issue 17, 2012. (**F**) is from Meints et al. [46], (**G**) is from Meints et al. [47], (**H**) is from Milrot et al. [48] and (**I**) is from Meints et al. [18]. All the figures are published with permission.

**Figure 2 viruses-12-00020-f002:**
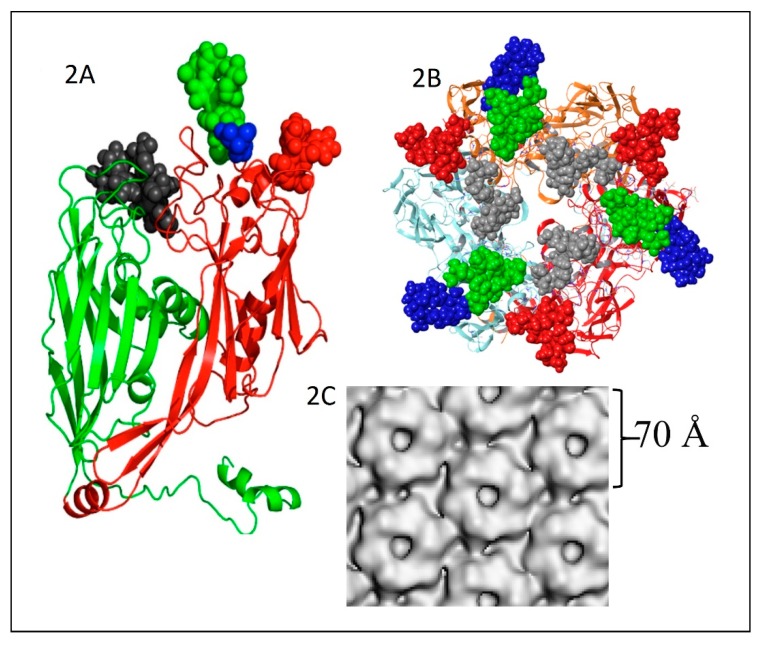
(**A**) Structure of a single PBCV-1 major capsid protein (Vp54). The four glycans attached to the major capsid protein at amino acids Asn 280 (green), Asn 302 (black), Asn 399 (red) and Asn 406 (blue) are on the outer surface of the virus. (**B**) Three of the major capsid proteins are assembled into a capsomer, viewed from the top. (**C**) Cryo-EM close up view of some capsomers. (**A**,**B**) are from De Castro et al. [51] and (**C**) is from Yan et al. [52]. All are published with permission.

**Figure 3 viruses-12-00020-f003:**
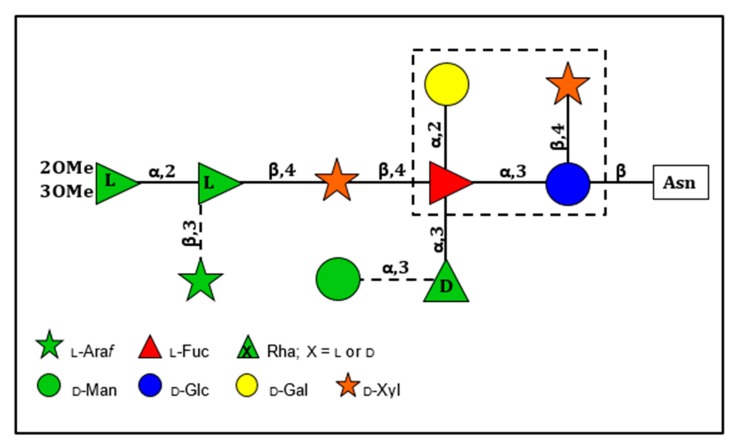
Structure of the glycan attached to the chlorovirus PBCV-1 major capsid protein (Vp54). Monosaccharides (mannose and arabinose) connected by dashed lines are not stoichoiometric substituents. The monosaccharides that are boxed are conserved in the glycans from all the chloroviruses. The figure is slightly modified from a figure from De Castro et al. [58].

**Figure 4 viruses-12-00020-f004:**
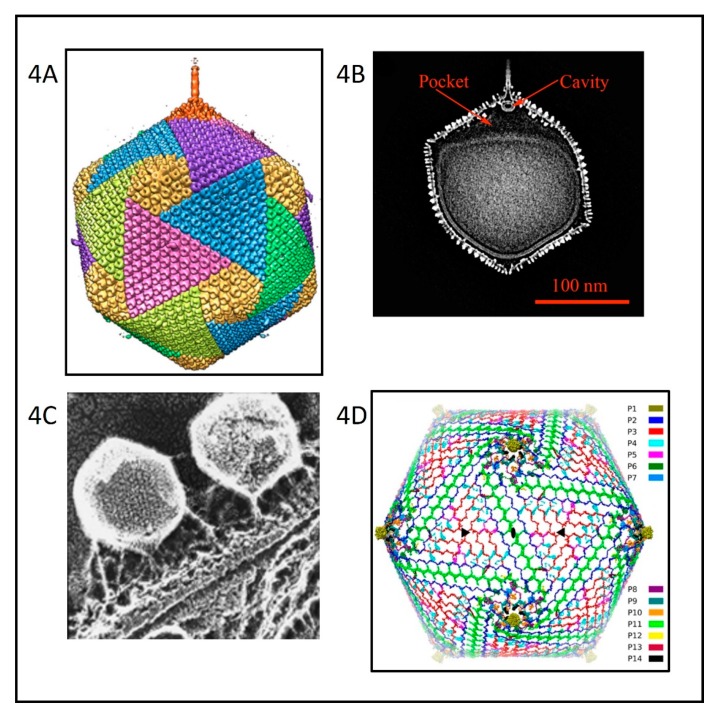
CryoEM structure of PBCV-1. (**A**) Hexagonal arrays of major capsomers form trisymmetrons and pentasymmetrons. The unique vertex with its spike structure is at the top. Capsomers in neighboring trisymmetrons are related by a 60° rotation, giving rise to the boundary between trisymmentrons. A fiber extents from one of the capsomers in each of the trisymmetrons. (**B**) Central cross-section of the cryo-EM density. A pocket is located between the virus internal membrane and the unique vertex and a cavity is located at the bottom of the spike structure. (**C**) PBCV-1 attached to the cell wall of its host chlorella as viewed by quick-freeze, deep etch microscopy. Note the virions are attached to the wall by fibers. (**D**) The cryoEM density (3.5 Å resolution) of PBCV-1 after removing the major capsid protein so that 14 minor proteins are visible. Each protein is shown in a different color as indicated on the right. (**A**,**B**) are from Cherrier et al. [61], (**C**) is from Van Etten et al. [27], and (**D**) is from Fang et al. [64]. All figures are published with permission.

**Figure 5 viruses-12-00020-f005:**
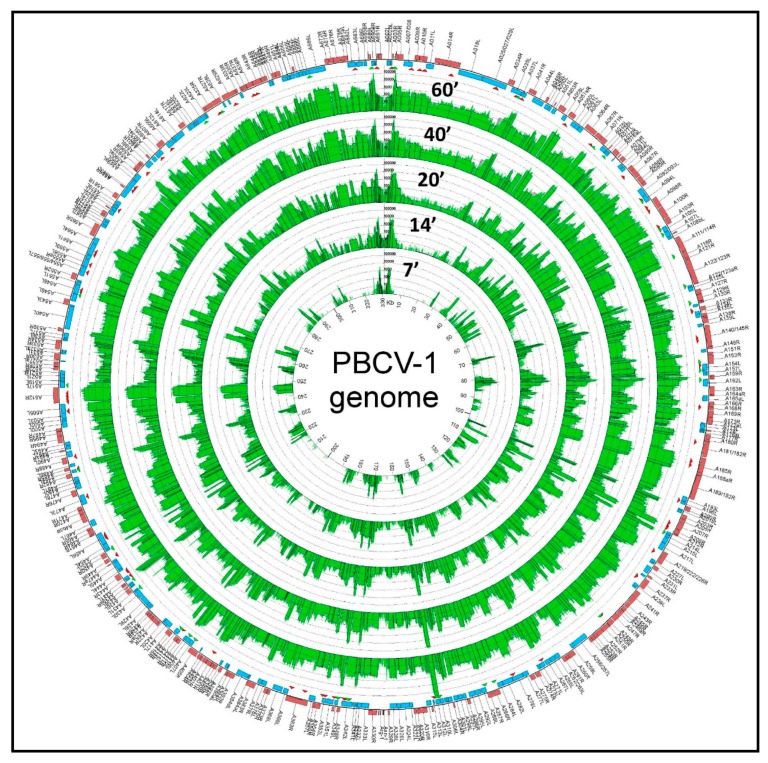
Mapping of the PBCV-1 transcriptome at various times post infection. PBCV-1 genes on the forward and reverse strands are depicted by red and blue boxes, respectively. The green curves in the interior concentric circles represent the normalized read coverage for each time point (7, 14, 20, 40, and 60 min p.i.) of the experiment in logarithmic scale (base 10). Note: the PBCV-1 genome is a linear molecule with inverted repeats and closed hairpin ends that is depicted as a circle. The two ends are at the 12 o’clock position. The figure is from Blanc et al. [75] and published with permission.

**Figure 6 viruses-12-00020-f006:**
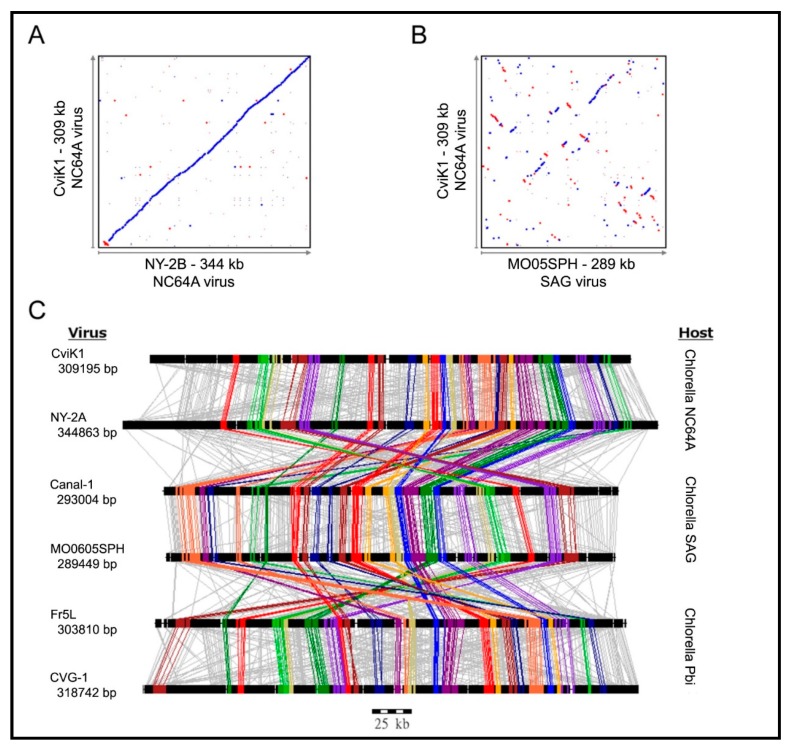
Gene dot-blots and gene alignments across chlorovirus host types. Dot-blot alignments of two chlorovirus genomes infecting the same (**A**) or different (**B**) hosts (blue indicates common genomic orientation, red indicates opposite genomic orientation). Each dot represents a protein match (ortholog) between the two viruses (BLASTp expect value > 1 × 10^−5^). (**C**) Schematic representation of six chlorovirus genomes from three different hosts. Each genome is depicted as a black line with genic regions represented by black boxes. Lines connect genes from the same family of homologous proteins (for clarity purpose, the six bigger families were removed from the figure). The colored set of genes and lines represents the Gene Gangs referred to in the text. While viruses that infect the same host demonstrate good conservation of synteny (**A**), synteny is poorly conserved across the virus types (**B**). Despite generally poor syntenic conservation, gene-centric alignments suggest that some conserved collinear blocks exist (**C**). Figure taken from Seitzer et al. [74] and published with permission.

**Figure 7 viruses-12-00020-f007:**
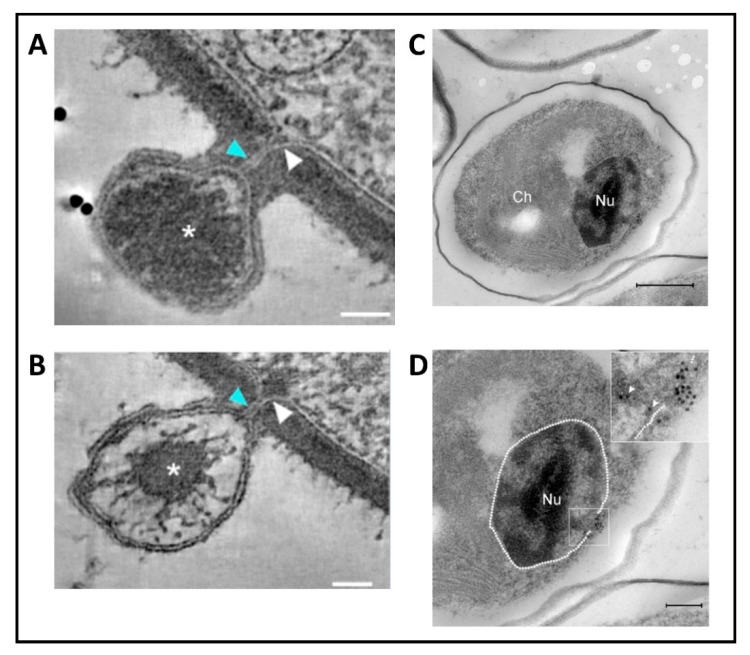
(**A**,**B**) PBCV-1 infected chlorella cells at 1.5–2 min p.i. were examined by Scanning-Transmission Electron Microscopy (STEM) tomography. The membrane-lined channel connecting the virus genome with the interior of the host is clearly visible. (**A**) A 7.8 nm tomographic slice from a 220 nm-thick STEM tomograph showing the close proximity between the viral and host internal membranes resulting from their convergence at the infection site. (**B**) A 5.2 nm tomographic slice from a different 220 nm STEM tomogram showing that part of the virus genome has been ejected into the cell. (**C**,**D**) Cells were infected with PBCV-1 for 6 min and then chemically fixed and thin sections were subjected to Electron Microscopy In Situ hybridization. (**C**) Low magnification of a cell illustrating dense viral DNA near the nucleus. (**D**) High magnification view of panel C. Note that viral DNA is probably entering the nucleus (white arrowheads in the inset). The nucleus contour is delineated with a white dashed line. The white and blue arrows (in A+B) point to the membranes that line the channel, the asterisk: viral DNA, Nu: nucleus, Ch; chloroplast, Scale bars: A, B: 50 nm; C: 500 nm; D: 200nm. Used with permission from Milrot et al. [113].

**Figure 8 viruses-12-00020-f008:**
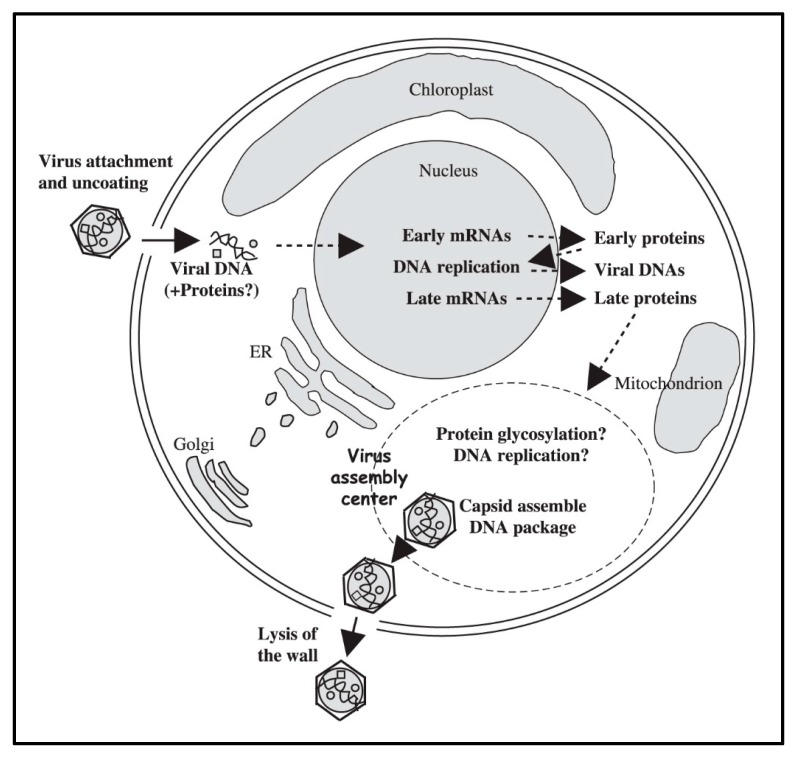
Proposed replication cycle of PBCV-1. The virus binds to the surface of the alga and opens a portal to release the viral DNA and associated proteins; the viral DNA moves to the nucleus where early gene transcription begins at 5 to 10 min p.i. Early mRNAs move to the cytoplasm for translation, and at least some early proteins presumably return to the viral genome located at or near to the nucleus to initiate viral DNA replication, which begins ~60 p.i., followed by late gene transcription. Late mRNAs move to the cytoplasm for translation and many of these late proteins are targeted to the virus assembly centers, where virus capsids are formed and DNA is packaged. Mature infectious virus particlles appear in the cytoplasm of the cell ~45 min prior to virus release. The chlorella cell membrane and wall lyses, and infectious PBCV-1 progeny viruses are released at 6 to 8 h p.i. (→) Known events; (---->) hypothesized events. Figure taken from Kang et al. [150] and used with permission.

**Figure 9 viruses-12-00020-f009:**
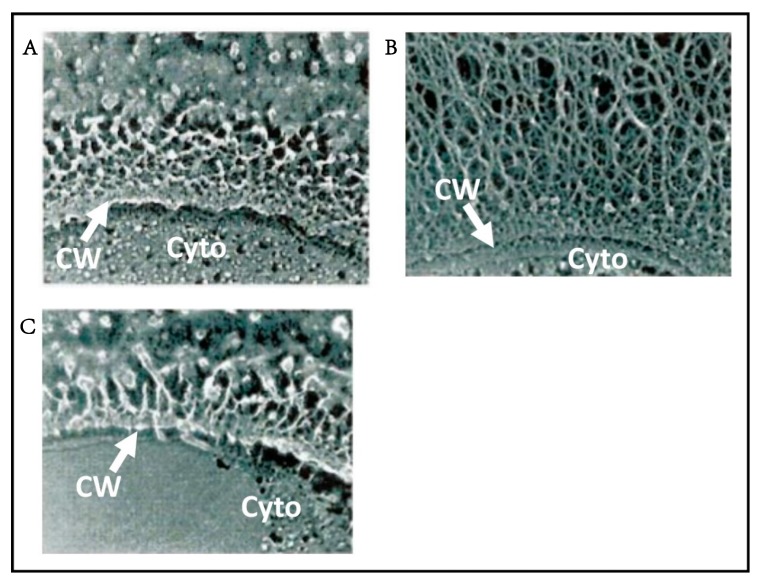
Hyaluronan on the surface of PBCV-1 infected *C. variabilis* NC64A. The figure shows cross-sections of (**A**) the surface of uninfected cells; (**B**) cells at 4 h p.i., and (**C**) cells at 4 h p.i. that were treated with hyaluronan lyase. Note: after treatment with hyaluronan lyase, the cell surface resembles the surface of uninfected cells. Chlorella cell wall varies from 50 to 100 nm in thickness [46,113]. CW is the cell wall and Cyto is the cytoplasm. Micrographs were taken from Graves et al. [204] with permission.

**Figure 10 viruses-12-00020-f010:**
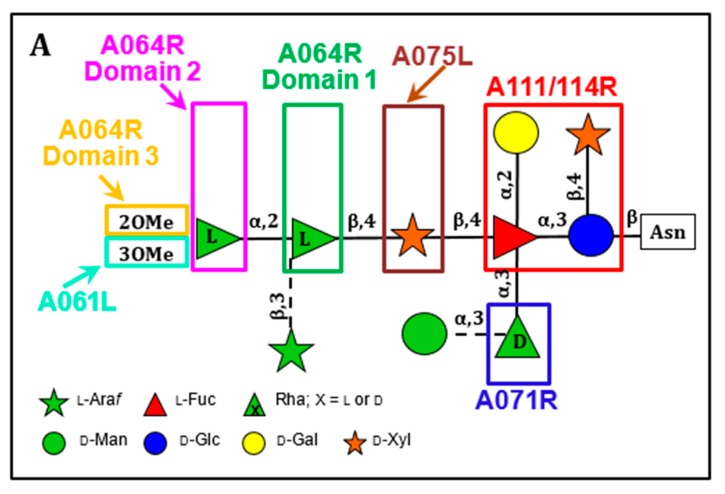
(**A**) PBCV-1 glycan structure with the predicted PBCV-1 encoded glycosyltransferases (named in the colored boxes) involved in the glycosidic bond formation. (**B**) Protein A064R has three domains with each domain carrying out a specific enzymatic function that has been verified biochemically: domain 1 is a β-l-rhamnosyltransferase that links l-rhamnose to d-xylose, domain 2 is an α-l-rhamnosyltransferase that links l-rhamnose to l-rhamnose, and domain 3 is a methyltransferase that places a methyl group at the 2 position on the terminal l-rhamnose [217].

**Figure 11 viruses-12-00020-f011:**
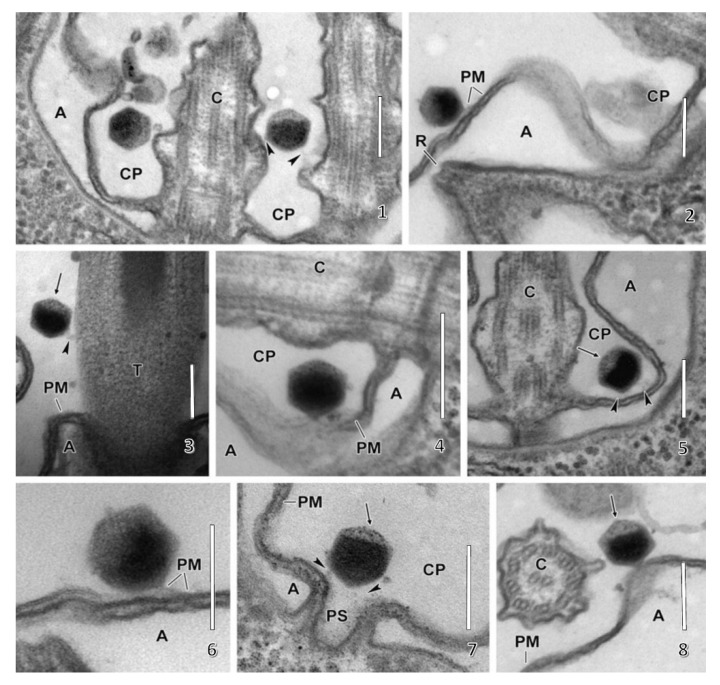
Electron microphotographs of the virus-containing *Paramecium bursaria–Chlorella* symbiotic system. The chloroviruses are associated with the somatic cortex of the ciliate. Arrowheads mark the tiny hair like fibers, connecting the virus particles to the paramecium plasma membrane. A unique electron-transparent virus vertex is shown with an arrow in Figure 1, Figure 4 and Figure 5. Longitudinal sections of the ciliary pits and virus particles are located in the tight space of a ciliary pit and between the cilia. The ciliary pits are on average only slightly wider than the diameter of the virus particles, and thus the viruses are shielded by the evaginating alveoli sacs. Viruses appear to be attached to the plasma membrane directly by the vertex opposite the unique electron-transparent vertex, or by one of the other vertices and via hair-like fibers. Therefore, the unique vertex of the virus is ready to attach to their host algae when the paramecia are disrupted. (**3**) Chlorovirus particles attached to the expulsive trichocyst. (**2**, **6**) Virus particles in association with the plasma membrane on the ciliary ridge. Note, that on the enlarged image (**6**) the side of the capsid appears very close to the membrane. (**7**) The viral capsid in the ciliary pits and near the entry of the parasomal sac (place of endocytosis), a virus particle attached to the membrane by the tiny hairlike fibers. (**8**) A virus particle attached to the plasma membrane of the surface and to the cilia. The virus is recognized by the electron dense nucleocapsid and a distinct electron transparent vertex pointed upwards and opposite the vertex attached to the ciliate plasma membrane. A—alveoli; C—cilia; CP—ciliary pits; PM—plasma membrane; PS—parasomal sac; T—trichocyst. Scale bars 200 nm. Published with permission from Yashchenko et al. [227].

**Figure 12 viruses-12-00020-f012:**
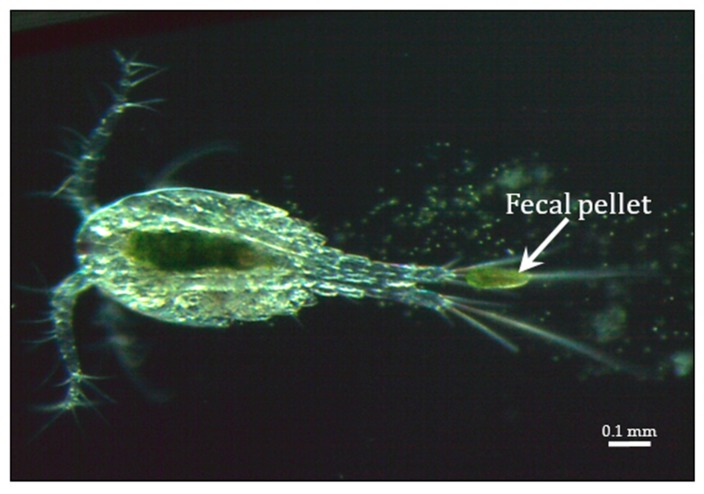
A copepod (*Eucyclops agilis*) gut full of disrupted *Paramecia bursaria* after foraging on the green paramecia and shown releasing a green fecal pellet. Incubating an isolated fecal pellet for 12 h resulted in an ~500-fold increase in chlorovirus titers.

**Figure 13 viruses-12-00020-f013:**
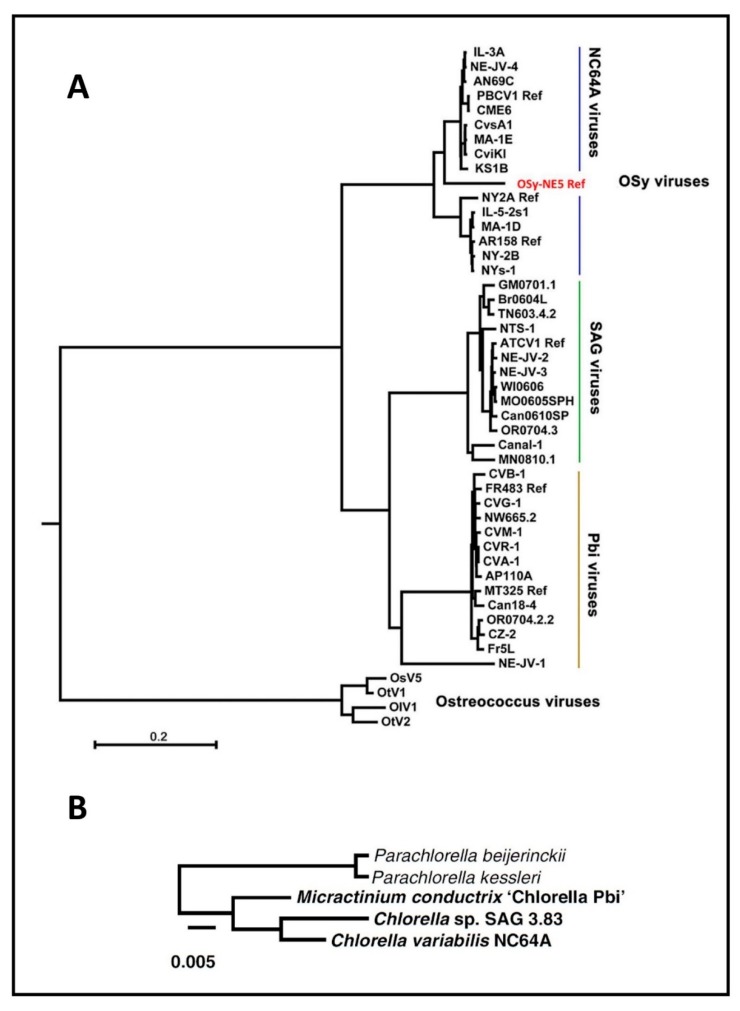
Phylogenies of chloroviruses and algal hosts. (**A**) Phylogenetic tree shows the evolutionary relationships between 47 chloroviruses concatenated amino acid sequences (7762 gap-free sites). The Maximum Likelihood tree was constructed using the MEGA 6.0 software with the Maximum Likelihood algorithm and default setting. The bar length represents 0.2 substitutions per amino acid site. A recently characterized Osy chlorovirus species is indicated in red and resides between the separate phylogenetic subclades of NC64A viruses; one subclade contains PBCV-1 while the other subclade contains chlorovirus NY-2A. Both subclades share almost perfect gene colinearity and they replicate in the same host. Branch support was estimated from 1000 bootstrap replicates. Four Ostreococcus virus sequences served as outgroups to root the tree. (**B**) Maximum Likelihood tree of three algal hosts based on 18S RNA alignment (2266 gap-free sites). The phylogenetic tree was computed using the GTR + G + I substitution model. All interior branches received maximal support (100%). Parachlorella species served as the outgroup. Panel **A** is from Quispe et al. [94] and panel **B** is from Jeanniard et al. [93], both are published with permission.

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
