# Peer review of "Chloroviruses"

_viruses, 2019, doi:10.3390/v12010020_

Round 1

Reviewer 1 Report

Despite my fear of sounding too congratulatory, this was the easiest manuscript I have reviewed. I can provide no substantive criticisms of this work which was lead by a scientist who is THE pioneer of algal virus and, perhaps even more broadly, giant virus research. Simply put, this is a fabulous review that fully captures the remarkable history of chlorovirus research and I believe it would be perfectly reasonable to publish it “as is” (a first in my record after contributing well over 200 reviews for various journals).

It is fitting that this manuscript is simply titled “Chloroviruses” because it provides a thoughtful, comprehensive review of more than 30 years of research on this fascinating group of viruses. The authors provide an excellent summary of literature on this topic while also providing insights into interesting unpublished results; presumably, these results weren’t published because they were ‘null’ results that weren’t considered interesting at the time the experiments were conducted. However, the authors have offered new insights based on current studies that, in hindsight, corroborate or highlight previous conclusions. As just one example, the authors explain how current studies of chlorovirus genes encoding enzymes for protein glycosylation explain past results when experiments were conducted with the goal of producing recombinant viruses (lines 949-953). Moreover, I especially appreciated that the authors were willing to present current work that occasionally refutes their own earlier conclusions. For example, early reports suggested that late transcripts were not polyadenylated, but subsequent studies demonstrated that these results were likely a methodological artifact that is only now apparent after the nature of certain membrane channels was determined, and it is now believed that chlorovirus late transcripts are polyadenylated (lines 632-639). In summary, this review goes beyond a mere summary of published works on chloroviruses by providing insights that could only come from the group who’s contributions have lead this amazing field of study for many years. I’ll look forward to using this published manuscript as a key reference and source of information on chloroviruses, and believe others will find this review equally valuable.

That said, I can offer a few very minor comments/suggestions:

Line 71: Indigenous seems like the wrong word in this context. Would “natural waters” be more appropriate?

Line 182: “Lose” should be revised to “loss”.

Line 256: Should “protrudes” be revised to “protrusion” or is this sentence trying to say something different?

Fig 6. The labels from the original figure for panel C are pixelated and nearly illegible. I suggest rewriting these labels using any simple graphic software to simply cover up the original labels.

Fig 7. The “dense viral DNA near the nucleus” isn’t clear in this panel and should be highlighted or pointed out.

Author Response

Reply to Editor's comments.

Sentences similar to previously published papers. All of these regions to our previously published papers.  We have edited many of these longer regions to make them slightly different.  However, in a few cases the regions only consisted of one or two sentences, and some of these short regions, we did not change.  I hope this is OK.

2. References to unpublished results and manuscripts in preparation. These were corrected as you suggested.  All of these references were from out lab.

Reply to comments from Referee 1.

We want to thank the reviewer for the nice comments.

Line 71: Indigenous seems like the wrong word in this context. Would “natural waters” be more appropriate?  changed to natural.

Line 182: “Lose” should be revised to “loss”. Changed

Line 256: Should “protrudes” be revised to “protrusion” or is this sentence trying to say something different? changed as suggested

Fig 6. The labels from the original figure for panel C are pixelated and nearly illegible. I suggest rewriting these labels using any simple graphic software to simply cover up the original labels. Changed as suggested

Fig 7. The “dense viral DNA near the nucleus” isn’t clear in this panel and should be highlighted or pointed out. The insert in Fig. 7D would seem to take care of this issue.

Reviewer 2 Report

This is a rather compete overview of the state of art in chlorovirus research, produced by a well-known group that has pioneered research in this field for many years, and published many articles in high-ranking journals. They have also produced review articles on this subject fairly regularly, (they do cite some of the more recent ones). The text is written in a story-like style that is easy to read and informative, but this does also mean that it’s rather long. In case space it at a premium, I made a few minor suggestions for shortening as I read through (below), and if required some details could be omitted on different points by using the citations (which are adequate), being more concise and by mentioning only the take-home messages rather than giving a more detailed explanation. However, it is a useful global overview and makes a good introduction for people not so familiar with the field.

Overall,  would suggest that this review is acceptable with some minor modifications and one extra table (see below).

In the abstract, the authors claim “and unlike all other viruses, enzymes involved in the glycan synthesis of the virus major capsid glycoproteins.” I’m not sure that this claim can now be made so strongly, although experimental verification of glycan synthesis clearly the most advanced for PBCV‑1. Other authors have now identified putatively similar kinds of genes in several other very large DNA viruses , as discussed by Piacente, F., Gaglianone, M., Laugieri, M.E., and Tonetti, M.G. (2015). The Autonomous Glycosylation of Large DNA Viruses. International Journal of Molecular Sciences 16, 29315–29328.

The authors might thus consider toning down the strength of their statement on this point.

At numerous places in the text, see for example pages 21–22, the authors often mention that chloroviruses encode the smallest examples of the proteins whose biochemical functions have been studied. While this was probably true at the dates if the original publications cited, I’m not sure that it is true for all of these proteins at present. I would thus suggest including a table comparing the lengths of these proteins with example representatives of other genera in the Phyconaviridae (Coccolithovirus, Phaeovirus, Prasinovirus, Prymnesiovirus, Raphidovirus), to show that claim is still true, or adjust their discussion on this point accordingly.

Unpublished results are mentioned at least 19 times during the text. While this is interesting for a reader, the authors should confirm to the editor that appropriate permissions have been granted for mentioning these results (unless they originate from the author’s laboratory).

Figure 11 needs some clarification. Numbering  of panels goes from 5 to 12. The panels need relabelling, preferably starting with “A” or “1”, and making the labels more distinct from the background. It’s not clear what the vertical bars above these labels represent. Are they scale bars? In any case, please show and mention the scale.

Detailed remarks

line

Article Text

Suggestion

Comment

75

"It should be noted that”

delete

83

“which is good reason to have zoochlorellae as a partner.

Could be omitted

84

In contrast, the reason the zoochlorellae would become a symbiont was unknown.

Could be omitted

100

Figure 1, B, C, D, E

Add scale size bars

More important for B, C since the size of the virus can also be seen in F

125

hr

h or hours

hr is not the SI unit.

It is interesting that

omit

206–7

However, it is now known that it exists as a trimer in the viral capsomers with a size of ~162 kD.

Could be omitted if the MW is put in line 203

281

“ER”

“endoplasmic reticulum”

Please spell out/explain abbreviations at first occurrences

309

Avoid splitting “PBCV‑1" on 2 lines (non-break hyphen required)

374–6

“Interestingly, poxviruses and African Swine Fever virus do something similar in regards to the gene deletions at one termini and replacement with a terminal sequence from the other end of the genome

Too colloquial – please rephrase

438, Fig. 6

Text too small

Increase font size in A, B,  especially in C.

Please also improve resolution in C (the lines are too grainy)

467

is ~40

is ~40%

496

It is interesting that

delete

563

hr

h or hours

hr is not the SI unit.

587

Avoid splitting “PBCV‑1" on 2 lines (non-break hyphen required)

863–4 Fig. 9

Please add scale bar sizes to TEM images.

969

Avoid splitting “PBCV‑1" on 2 lines (non-break hyphen required)

972

Arabitopsis

Arabidopsis

990

methy group

methyl group

1057

There is one report that explored the possibility that

delete

1286

This is reflected in that

delete

1291

However, it is also true that

delete

1304

although one is always continuing to try and develop such systems.

delete

Author Response

Overall,  would suggest that this review is acceptable with some minor modifications and one extra table (see below).

In the abstract, the authors claim “and unlike all other viruses, enzymes involved in the glycan synthesis of the virus major capsid glycoproteins.” I’m not sure that this claim can now be made so strongly, although experimental verification of glycan synthesis clearly the most advanced for PBCV‑1. Other authors have now identified putatively similar kinds of genes in several other very large DNA viruses , as discussed by Piacente, F., Gaglianone, M., Laugieri, M.E., and Tonetti, M.G. (2015). The Autonomous Glycosylation of Large DNA Viruses. International Journal of Molecular Sciences 16, 29315–29328.

The authors might thus consider toning down the strength of their statement on this point.

"Unlike all other viruses" was removed from the sentence.

At numerous places in the text, see for example pages 21–22, the authors often mention that chloroviruses encode the smallest examples of the proteins whose biochemical functions have been studied. While this was probably true at the dates if the original publications cited, I’m not sure that it is true for all of these proteins at present. I would thus suggest including a table comparing the lengths of these proteins with example representatives of other genera in the Phyconaviridae (Coccolithovirus, Phaeovirus, Prasinovirus, Prymnesiovirus, Raphidovirus), to show that claim is still true, or adjust their discussion on this point accordingly.

We have rechecked the sizes of the proteins mentioned and this still holds for the Type II DNA topoisomerase, the histone H3K27 methylase and the mRNA capping enzyme.  This is not true for the others in the text and we have changed smallest enzyme to among the smallest enzymes. We do not seen any reason to include a table listing the size of these enzymes.

Unpublished results are mentioned at least 19 times during the text. While this is interesting for a reader, the authors should confirm to the editor that appropriate permissions have been granted for mentioning these results (unless they originate from the author’s laboratory).

All of the unpublished results are from the authors lab so this is not an issue.

Figure 11 needs some clarification. Numbering  of panels goes from 5 to 12. The panels need relabelling, preferably starting with “A” or “1”, and making the labels more distinct from the background. It’s not clear what the vertical bars above these labels represent. Are they scale bars? In any case, please show and mention the scale.

The figure numbers have been corrected and the of the scale bars are now listed in the legend to the figure.

Detailed remarks

line

Article Text

Suggestion

Comment

75

"It should be noted that”

delete

Deleted

83

“which is good reason to have zoochlorellae as a partner.

Could be omitted We prefer to keep this sentence in the manuscript.

84

In contrast, the reason the zoochlorellae would become a symbiont was unknown.

Could be omitted. We prefer to keep this sentence in the manuscript.

100

Figure 1, B, C, D, E

Add scale size bars

More important for B, C since the size of the virus can also be seen in F. This was done.

125

hr

h or hours

hr is not the SI unit. hr changed to h as suggested.

It is interesting that

omit

Changed as suggested

206–7

However, it is now known that it exists as a trimer in the viral capsomers with a size of ~162 kD.

Could be omitted if the MW is put in line 203. Want to keep the sentence as is.

281

“ER”

“endoplasmic reticulum”

Please spell out/explain abbreviations at first occurrences. Changed this as suggested.

309

Avoid splitting “PBCV‑1" on 2 lines (non-break hyphen required). Done

374–6

“Interestingly, poxviruses and African Swine Fever virus do something similar in regards to the gene deletions at one termini and replacement with a terminal sequence from the other end of the genome

Too colloquial – please rephrase. Prefer to keep it as is.

438, Fig. 6

Text too small

Increase font size in A, B,  especially in C.

Please also improve resolution in C (the lines are too grainy)

467

is ~40

is ~40%

changed

496

It is interesting that

delete

563

hr

h or hours

hr is not the SI unit.

Changed as recommended

587

Avoid splitting “PBCV‑1" on 2 lines (non-break hyphen required)

changed accordingly

863–4 Fig. 9

Please add scale bar sizes to TEM images.

Did it.

969

Avoid splitting “PBCV‑1" on 2 lines (non-break hyphen required)

Changed accordingly

972

Arabitopsis

Arabidopsis

Changed

990

methy group

methyl group

Changed

1057

There is one report that explored the possibility that

delete

Changed to "One report explored...."

1286

This is reflected in that

delete

prefer to keep it as is.

1291

However, it is also true that

delete

Changed

1304

although one is always continuing to try and develop such systems.

delete                                 Changed to "One continues to try and ..."